# Inter-annual variation in seasonal dengue epidemics driven by multiple interacting factors in Guangzhou, China

Rachel J. Oidtman[1], Shengjie Lai [2,3,4], Zhoujie Huang[2], Juan Yang[2], Amir S. Siraj [1], Robert C. Reiner Jr.[5], Andrew J. Tatem[3,4], T. Alex Perkins[1] & Hongjie Yu[2]

Vector-borne diseases display wide inter-annual variation in seasonal epidemic size due to their complex dependence on temporally variable environmental conditions and other factors. In 2014, Guangzhou, China experienced its worst dengue epidemic on record, with incidence exceeding the historical average by two orders of magnitude. To disentangle contributions from multiple factors to inter-annual variation in epidemic size, we fitted a semi-mechanistic model to time series data from 2005–2015 and performed a series of factorial simulation experiments in which seasonal epidemics were simulated under all combinations of year-specific patterns of four time-varying factors: imported cases, mosquito density, temperature, and residual variation in local conditions not explicitly represented in the model. Our results indicate that while epidemics in most years were limited by unfavorable conditions with respect to one or more factors, the epidemic in 2014 was made possible by the combination of favorable conditions for all factors considered in our analysis.

[1] Department of Biological Sciences and Eck Institute for Global Health, University of Notre Dame, Notre Dame 46556 IN, USA. [2] School of Public Health, Fudan University, Key Laboratory of Public Health Safety, Ministry of Education, Shanghai 200032, China. [3] WorldPop, Department of Geography and Environment, University of Southampton, Southampton SO17 1BJ, UK. [4] Flowminder Foundation, Stockholm SE-11355, Sweden. [5] Institute for Health and Metrics and Evaluation, University of Washington, Seattle 98195 WA, USA. Correspondence and requests for materials should be addressed to T.A.P. (email: taperkins@nd.edu) or to H.Y. (email: yhj@fudan.edu.cn)

In response to warming temperatures and other consequences of climate change, many regions are undergoing changes in their suitability for pathogens whose transmission is sensitive to weather conditions[1,2]. Although there are clear links between local weather conditions and the transmission of numerous pathogens undergoing changes in response to climate change, those links can be difficult to isolate in epidemiological analyses[3,4]. In an area where a pathogen is endemic, its host population may often be considerably immune, resulting in inter-annual variation driven by a combination of time-varying weather conditions and nonlinear feedbacks of population immunity[5–7]. Those feedbacks may furthermore vary with shifting patterns of host demography, which modulate ebbs and flows in the pool of susceptible hosts over time[8]. In an area where a pathogen is not endemic but is instead transmitted in the context of limited seasonal epidemics, pathogen importation can play a critical role in limiting or enabling transmission, depending on whether importation occurs at times when weather conditions are conducive to local transmission[9].

One disease that is subject to this entire suite of issues is dengue. This viral disease is vectored by *Aedes aegypti* and *Aedes albopictus* mosquitoes, which have a life cycle that is sensitive to both temperature[10] and the availability of water, either from rainfall[11] or human-associated water resources[12]. The time required for dengue virus (DENV) to incubate in the mosquito (known as the extrinsic incubation period) is also highly sensitive to temperature[10], which in turn affects the proportion of adult female mosquitoes that live long enough to become infectious to humans[13]. At the same time, each of the four DENV serotypes confers lifelong homotypic immunity, which can result in a significant dampening of transmission in endemic settings[7]. These issues, along with complex differences and interactions among its serotypes, heterogeneous rates of reporting, and local differences in human living conditions have made it challenging to isolate the influence of weather conditions on DENV transmission[14,15]. Low population immunity could make associations between climatic and weather conditions and DENV transmission clearer in low-transmission settings, but importation often dominates incidence patterns in settings with low population immunity to such a degree that opportunities to investigate the influence of weather conditions on local transmission can be somewhat limited[16].

The recent history of DENV in mainland China presents an ideal opportunity to examine how temporal variation in local climatic conditions and pathogen importation interact to drive inter-annual variability in transmission in a seasonally epidemic context. Since 1990, mainland China has experienced highly variable, but relatively low DENV transmission, with a median of 376 and a range of 2–6836 cases reported annually from 1990 to 2004[17]. More recently, increasingly large seasonal epidemics have occurred, with a median of 438 and a range of 59–47,056 cases from 2005 to 2014[17]. These epidemics have been highly seasonal and distinct from year to year, given the markedly seasonal climatic conditions in portions of mainland China where dengue is locally transmitted. At the same time, the endemic status of DENV in neighboring southeast Asia ensures a reliable, and growing, source of DENV importation[18].

Of central importance to recent trends for dengue in mainland China is Guangzhou, a city of 14 million in the southern province of Guangdong, where 94.3% of all locally acquired dengue cases in mainland China occurred during 2005–2014[17]. Following an epidemic of 37,445 locally acquired dengue cases in Guangzhou in 2014, there has been growing interest in modeling this event. Different models have identified different drivers, leading to inconsistent conclusions. Two studies concluded that weather conditions were the primary driver of DENV transmission[19,20], whereas others concluded that importation patterns, delayed

outbreak response, or both importation patterns and delayed outbreak response were causal drivers of the 2014 epidemic[21–23]. Still others found that neither weather conditions nor importation was key drivers of transmission, but instead that urbanization was pivotal[24,25]. Two analyses[19,20] that used incidence data aggregated at a monthly timescale for 2005–2015 showed high predictive capability at one-month lead times but did not facilitate clear interpretation of how importation interacts with local conditions to result in high inter-annual variation in transmission. Mechanistic models applied to date have used daily or weekly data, but only for 2013–2014, and therefore only considered years with anomalously high transmission[21,23–25]. As a result, it is unclear how well those models could explain the strikingly low incidence observed in years other than 2013–2014.

Here, we applied a stochastic, time-series susceptible-infected-recovered (TSIR) model[26] that we fitted to daily dengue incidence data from 2005 to 2015 to determine the relative roles of local conditions and pathogen importation in driving inter-annual variation in DENV transmission. To make detailed inferences with incidence data at daily resolution, we made several enhancements to the standard TSIR framework, including a realistic description of the DENV generation interval, lagged effects of covariates on transmission, and flexible spline relationships between covariates and their contributions to transmission (Fig. 2). After fitting the model and checking its consistency with the data to which it was fitted, we conducted simulation experiments in which we examined how the annual incidence of locally acquired dengue differed across simulations with inputs about local conditions and importation patterns that varied from year to year. For the purpose of these analyses, we considered local conditions to be those that relate to the term describing local transmission in our model, which include mosquito density, temperature, and other unspecified factors captured by a time-varying residual term. Using a series of factorial simulation experiments, we quantified the relative contributions of local conditions and importation to inter-annual variation in dengue incidence.

## Results

**Relationships between local conditions and transmission.** In the 11-year time series that we examined, there was marked seasonal variation in local dengue incidence and in putative drivers of DENV transmission (Fig. 1). We estimated a latent mosquito density curve, $m(t)$, using two different types of entomological data (Breteau index, mosquito ovitrap index) (Fig. 1c), resulting in a seasonal pattern of mosquito density. Although inter-annual variation in mosquito density and temperature was minimal (Fig. 1c, d), there was pronounced inter-annual variation in imported and local dengue incidence (Fig. 1a, b).

To understand the association between temporal variation in local conditions and local DENV transmission, we fitted two bivariate basis functions with cubic B-splines that allowed for distinct lagged relationships between temperature, mosquito density, and their effects on the transmission (Fig. 2). For temperature, we found that its contribution on a given day to the time-varying transmission coefficient, $\beta(t)$, peaked near the temperature optimum of 33.3 °C assumed by the prior but with considerable uncertainty (posterior median: 33.6 °C; 95%: 24.8–36 °C). The magnitude of these daily temperature contributions was somewhat larger at shorter lags (Fig. 2a). For mosquito density, we found a positive relationship between $m(t)$ and $\beta(t)$ at all lags, with the daily contribution of mosquito density at intermediate (~20–30 d) lags being 10–15% larger than at shorter (~1–20 d) or longer (~30–50 d) lags (Fig. 2b). In addition, we estimated a term, $\beta_0(t)$, that explicitly modeled residual variation

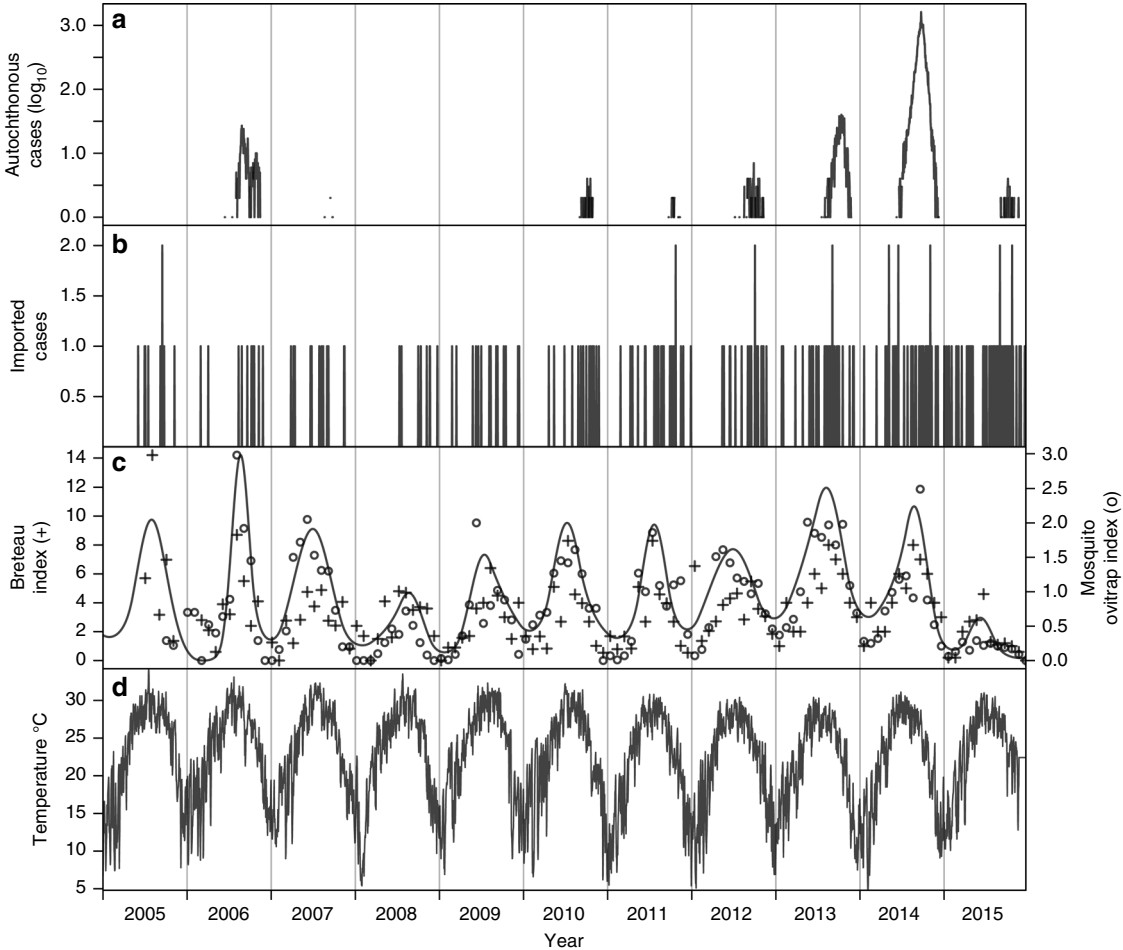

**Fig. 1** Time series of data from 2005 to 2015 in Guangzhou, China. **a** Local dengue incidence. **b** Imported dengue incidence. **c** Breteau index (BI) of mosquito density indicated by '+', mosquito ovitrap index (MOI) of mosquito larval density indicated by 'o', and maximum-likelihood estimate of the latent mosquito density variable, $m(t)$, indicated by curve. **d** Daily mean temperature

in $\beta(t)$ that was not accounted for by temperature or mosquito density but that was necessary to reproduce observed patterns of local dengue incidence (Fig. 3). The 95% posterior predictive interval for this term was entirely positive during the transmission season in 2014, whereas in other years it fluctuated relatively tightly around zero (Fig. 3b). Under an alternative model formulation with a constant residual variation term (Supplementary Methods 2), the model could not match the observed patterns of local incidence; specifically, this alternative model formulation could not recreate the unusually high incidence observed in 2014 (Supplementary Methods 2 Fig. 10). This implies that appealing to systematic differences with respect to one or more unspecified local conditions is necessary to explain the anomalously high incidence observed in 2014.

By estimating three separate components of $\beta(t)$, we were able to evaluate the relative contributions to $\beta(t)$ of each of mosquito density, temperature, and unspecified local conditions at different points in time (Fig. 3). We found that mosquito density tended to have a smaller but more variable effect compared to temperature, which resulted in considerably lowered $\beta(t)$ values at low temperatures. In most years, the effect of mosquito density on $\beta(t)$ tended to be more pronounced within a shorter seasonal time window than did that of temperature, as indicated by peaked patterns of contributions of mosquito density to transmission (Fig. 1c) relative to broad, flat patterns of contributions of temperature to transmission (Fig. 3d). The contributions of $\beta_0(t)$ to $\beta(t)$ were highly variable across different draws from the

posterior, other than the consistently large, positive effect in 2014 (Fig. 3b).

**Checking model consistency with data.** We used data on imported cases to seed 1000 simulations of local DENV transmission over the entire 2005–2015 time period, with local transmission patterns in each simulation determined by a different random draw from the posterior distribution of model parameters. Over the period as a whole, daily medians of simulated local dengue incidence were highly correlated ($\rho = 0.966$) with observed local incidence (Fig. 4a). Within each year, observed features of local dengue incidence patterns were generally consistent with simulations. For annual incidence across years and peak weekly incidence, observed values fell within the 95% posterior predictive intervals (PPI) in 11/11 years (Supplementary Figs. 1, 2). For total number of weeks with non-zero incidence and for the length of the transmission season (time elapsed between first and last cases), observed values fell within the 95% PPIs in 8/11 years (Supplementary Figs. 3, 4). Years for which observed values fell outside of the 95% PPI tended to be those with intermediate levels of transmission (2006, 2013) or longer transmission seasons (2010, 2015). Years for which observed values most consistently fell within the 95% PPIs were those with either low (2007, 2008) or high (2014) transmission. In addition, we found that the fitted model correctly ranked 2014 as the year with the highest annual incidence 100% of the time

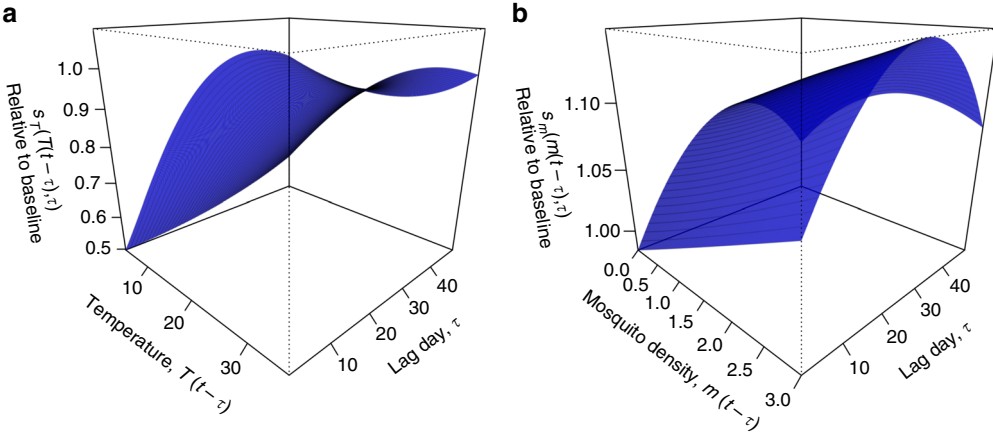

**Fig. 2** Relative contributions of temperature and mosquito density to the transmission coefficient on day t. Following Eq. (4), the surface in **a** is for temperature and corresponds to $s_T(T_{t-\tau}, \tau)$ and the surface in **b** is for mosquito density and corresponds to $s_m(m(t-\tau), \tau)$, both of which are summed across values of $\tau$ ranging 1–49 lag days, exponentiated, and multiplied by each other and $e^{\beta_0(t)}$ to obtain $\beta(t)$. Values of parameters informing these surfaces shown here represent medians from the posterior distribution of parameters

**Fig. 3** Posterior estimates of transmission coefficient and contributions thereto from fitted components. **a** Time-varying transmission coefficient, $\beta(t)$. Contributions to $\beta(t)$ from **b** residual local conditions, $\beta_0(t)$, **c** mosquito density, $m(t)$, and **d** temperature. Different colored lines correspond to different samples from the posterior distribution of parameter values, which provide information about correlations among parameters that pertain to different components of the model. The red horizontal line in **a** indicates $\beta(t) = 1$. The shaded blue region represents the 95% posterior predictive interval, and the black line is the median value

(Fig. 5). Other notable high years included 2013 and 2006, which were both correctly ranked as years with relatively high local incidence. Years with low local incidence that our model ranked

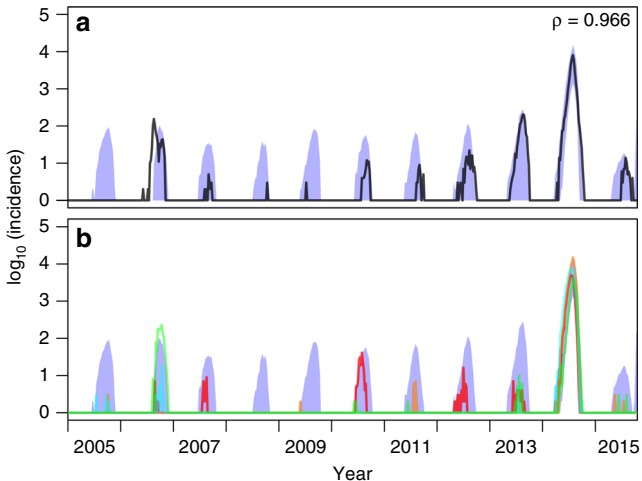

**Fig. 4** Correspondence between empirical and simulated patterns of local dengue incidence. In **a** the black line shows empirical values of log local incidence and the blue band shows the 95% posterior predictive interval from model simulations. The value of Pearson's correlation coefficient indicated in the upper right pertains to untransformed daily values between model simulations and data. In **b** different colored lines correspond to simulations based on different samples from the posterior distribution of parameter values. Simulations of local transmission were seeded by data on imported cases and otherwise used the fitted model of local transmission to simulate local cases

correctly included 2008–2011 and 2015. Altogether, these results suggest that our model is capable of recreating key features of the incidence time series and, therefore, is suitable for exploring drivers of inter-annual variation in local incidence. As described in Supplementary Methods 1, a simpler alternative model with many of the same features was found to be unsuitable for this task, underscoring the need for the level of detail incorporated into our primary model. Likewise, the inclusion of additional weather variables did not improve the model's ability to explain inter-annual variation in local dengue incidence (Supplementary Methods 2).

**Disentangling drivers of inter-annual variation**. Once we determined that simulations from the fitted model were consistent with observed patterns, we performed a two-way factorial simulation experiment in which we swapped local conditions (i.e., mosquito density, temperature, and residual variation in local transmission captured by $\beta_0(t)$) from each year with imported case patterns from each other year and used those conditions to drive simulations of 1000 replicate transmission seasons under each of these 122 combinations (Supplementary Fig. 5). Some of the more extreme contrasts illustrate the reasoning behind this approach. For example, given local conditions in 2014, our model projects that much higher local incidence would have been observed under importation conditions experienced in most other years (Fig. 6, top). In contrast, given imported case patterns from 2008, our model projects that very low local incidence would have resulted from local conditions in every year, including 2014 (Fig. 6, bottom). While simulations from these example years demonstrate how sensitive local incidence can be to interactions between importation and local conditions, the two-way analysis

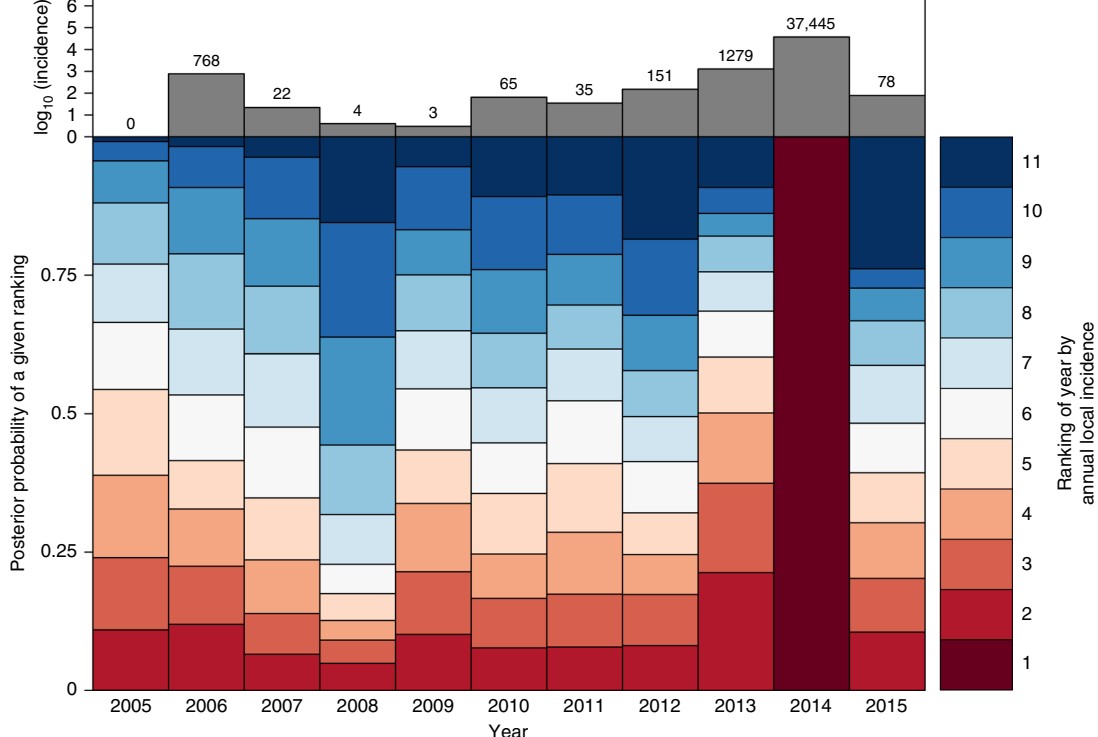

**Fig. 5** Ranking of years by simulated local annual incidence. Simulations were from the fitted model given data on local conditions and imported cases from that year. Dark red corresponds to the lowest ranking (i.e., highest simulated local incidence), and dark blue corresponds to the highest ranking (i.e., lowest simulated local incidence). The height of a given segment of a given year's bar is proportional to the posterior probability that simulations of local incidence from that year were of a given rank relative to other years. For example, the dark red segments in 2013 and 2014 indicate a high posterior probability that model simulations correctly resulted in high rankings for those years. The total number of dengue cases reported in each year is shown in the top panel for reference

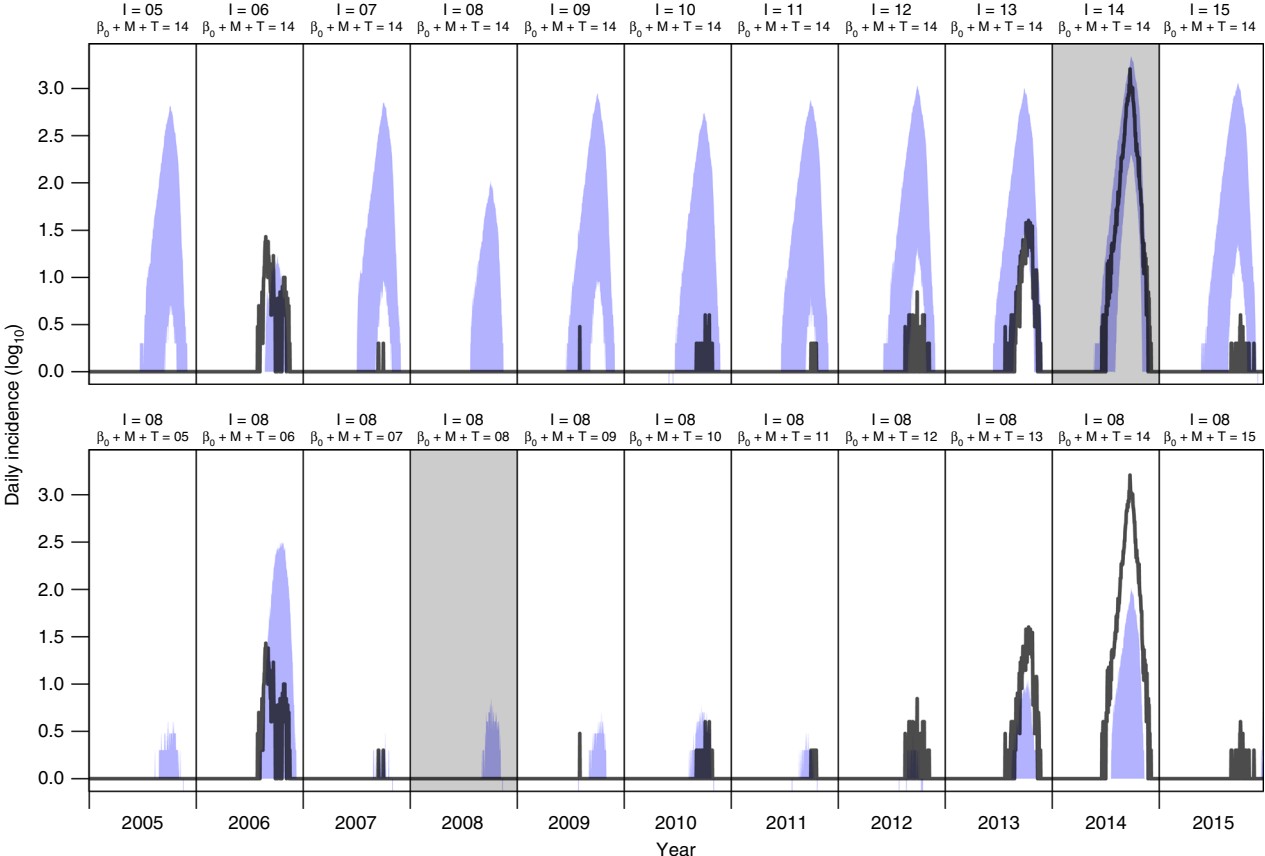

**Fig. 6** Example outputs from the factorial simulation experiments. The top row shows a series of 1000 simulations per year (blue: 95% posterior predictive interval) combining local conditions from 2014 (the year with highest local transmission) with importation patterns from each year in 2005–2015. The bottom row shows a series of 1000 simulations per year combining importation patterns from 2008 (the year with the lowest importation) with local conditions from each year in 2005–2015. These examples contrast seasonal epidemic patterns that would have resulted from hypothetical situations swapping year-specific local conditions and year-specific importation patterns from different years. For reference, empirical patterns of local incidence are shown with black lines. Similar simulations from all combinations of years of local conditions and importation patterns were taken into account in the two-way analysis of variance

of variance provides a summary of these interactions across all years considered in our study.

Of the 50.2% of variation in simulated local incidence that was accounted for by the model in a two-way analysis of variance, local conditions accounted for approximately 88.9% and imported case patterns accounted for the remaining 11.1% (Table 1, first row). A large amount of residual variation (49.8%) in simulated local incidence across replicates was not explained by either factor, which is consistent with the highly stochastic nature of epidemics in this setting where DENV transmission is so volatile. On the whole, these results showed that high local incidence was unlikely to occur in years in which local conditions were not highly suitable for transmission (Fig. 7). Performing a similar factorial simulation experiment in which we omitted 2014, we found that the model accounted for only 11.1% of the overall variation in simulated local incidence, 38.3% of which was accounted for by local conditions (Table 1, second row). This result showed that when we omitted 2014, the model accounted for much less variation in simulated local incidence than when 2014 was included and that importation patterns described more of the variation in simulated local incidence than local conditions.

To parse the individual contributions of each local variable to inter-annual variation in local incidence, we performed a four-way factorial simulation experiment in which we swapped all possible combinations of imported case patterns, mosquito density, temperature, and $\beta_0$ from different years. We performed

**Table 1 Sum of squared error from two-way factorial simulation experiment**

| Years included | Sum of squared error, SSQ (% Total, % Model) | | | |
|---|---|---|---|---|
| | $\beta_0 + M + T$ | $I$ | Residual | Total |
| 2005–2015 | $4.1 \times 10^5$ (44.6, 88.9) | $5.1 \times 10^4$ (5.6, 11.1) | $4.6 \times 10^5$ (49.8) | $9.1 \times 10^5$ |
| 2005–2013, 2015 | $1.7 \times 10^4$ (4.3, 38.3) | $2.8 \times 10^4$ (6.8, 61.7) | $3.7 \times 10^5$ (88.9) | $4.1 \times 10^5$ |

SSQ is attributable to inter-annual variation in local conditions ($\beta_0 + M + T$), imported case patterns ($I$), and residual variation from stochasticity across model simulations (Residual). The % Total values were calculated by dividing the SSQ explained by a given variable by the SSQ Total. The % Model values were calculated by dividing the SSQ explained by a given variable by the sum of SSQ values from the two model terms; thus, there is no % Model value to report in the Residual column

a set of 1000 replicate simulations for each of the 14,641 possible ways that year-specific patterns could be combined, allowing us to account for possible interactions among these four variables. Calculating the variation explained by $\beta_0$ in the four-way analysis of variance, we found that it accounted for 75.4% of all variation in local incidence accounted for by the model (Table 2). Imported cases contributed the next largest portion of variation (11.3%), whereas mosquito density (9.5%) and temperature (3.8%) each contributed less. Repeating this four-way analysis of variance without data from 2014, the proportions of all variation explained

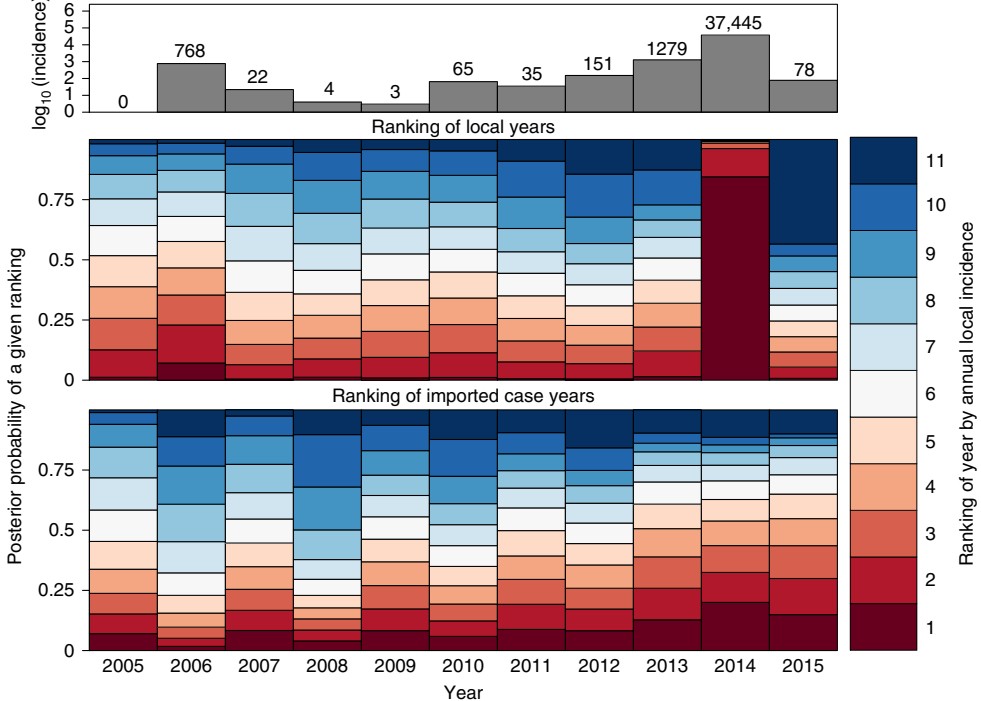

**Fig. 7** Ranking of all years by simulated annual incidence from the factorial simulation experiment. (Top) The total number of dengue cases reported in each year is shown in the top panel for reference. (Middle) Simulations with the fitted model given data on local conditions from a given year (each column) and imported cases from each of the other years. (Bottom) Simulations with the fitted model given data on imported cases from a given year (each column) and local conditions from each of the other years. A large amount of red in a column indicates that conditions in that year were relatively favorable for transmission, whereas blue indicates the opposite, similar to Fig. 5

| Table 2 Sum of squared error from four-way factorial simulation experiment | | | | | | |
|---|---|---|---|---|---|---|
| **Years included** | **Sum of squared error, SSQ (% Total, % Model)** | | | | | |
| | $\beta_0$ | $M$ | $I$ | $T$ | **Residual** | **Total** |
| 2005–2015 | $4.2 \times 10^7$ (31.1, 75.4) | $5.2 \times 10^6$ (3.9, 9.5) | $6.3 \times 10^6$ (4.6, 11.3) | $2.1 \times 10^6$ (1.6, 3.8) | $7.8 \times 10^7$ (58.7) | $1.3 \times 10^8$ |
| 2005–2013, 2015 | $7.6 \times 10^5$ (1.3, 10.5) | $2.6 \times 10^6$ (4.6, 35.9) | $2.9 \times 10^6$ (5.1, 39.2) | $1.0 \times 10^6$ (1.8, 14.3) | $5.0 \times 10^7$ (87.2) | $5.7 \times 10^7$ |

SSQ is attributable to inter-annual variation in each model term ($\beta_0$, $M$, $I$, $T$) and residual variation from stochasticity across model simulations (Residual). The % Total values were calculated by dividing the SSQ explained by a given variable by the SSQ Total. The % Model values were calculated by dividing the SSQ explained by a given variable by the sum of SSQ values from the four model terms; thus, there is no % Model value to report in the Residual column

by the models were somewhat more consistent across these four variables (Table 2, second row). Furthermore, the rankings of relative contributions of the four variables changed, with mosquito density (35.9%) and imported cases (39.2%) playing dominant roles in driving inter-annual variation in local incidence in the absence of data from 2014.

## Discussion

Populations subject to seasonal epidemics of any number of diseases are prone to high variability in epidemic size, due to inter-annual variation in imported cases that seed those epidemics and inter-annual variation in local conditions that drive transmission. We estimated the relative contributions of local conditions (i.e., temperature, mosquito density, and other unknown local factors) and importation in driving inter-annual variation in dengue epidemics in Guangzhou, China, which has recently been subject to seasonal epidemics ranging four orders of magnitude in size. Other studies[19–24] have investigated the same 11-year time series, either in whole or in part, but arrived at differing conclusions and did not take full advantage of the

exceptional level of detail in this data set (Supplementary Table 1). By leveraging these data more fully and using a modeling framework that blends elements of mechanistic and statistical modeling, we showed that local conditions and importation patterns jointly determined epidemic size in most years and that anomalies in unexplained conditions affecting local transmission were responsible for one anomalously large epidemic. Specific examples from this 11-year time series reinforce the notion that either or both of these factors can limit epidemic size.

Regarding the large epidemic in 2014, our results suggest that unknown factors captured by $\beta_0(t)$ played a dominant role in driving this extreme event. For this reason, the unknown factors that $\beta_0(t)$ was picking up on should now be of great interest. One possibility is that transmission was actually not much higher in 2014 but instead that a larger proportion of DENV infections resulted in symptomatic disease in 2014 than in other years. This could have occurred if a large number of residents of Guangzhou experienced a mild or asymptomatic first infection prior to 2014 and then experienced a more severe second infection in 2014[27]. This may be a possibility given that an increase in the diversity of

DENV serotypes circulating in this region was observed over the period of our study. Specifically, DENV-1 was the only serotype isolated from locally acquired cases in 2005–2008, and, while it remained dominant thereafter, the three other serotypes were also isolated in those years[28,29]. A second, related possibility is that there were serotype differences in the severity of symptoms[30]. However, an analysis[31] pairing information about serotype etiology and symptoms in a subset of cases did not support this hypothesis. A third possibility is that media attention during the 2014 epidemic[32,33] could have heightened awareness of dengue and led to an increase in the number of people detected by the surveillance system[34]. In summary, all of these hypotheses predict that some of the increase in reported cases in 2014 could have been attributable to an increase in the proportion of infections that resulted in detection by public health surveillance, albeit for different reasons.

It is also possible that high estimates of $\beta_0(t)$ for 2014 could reflect an increase in local transmission, rather than a difference in detectability. First, although mosquito densities were not notably higher in 2014 than in other years, there could have been undetected changes in the composition of the mosquito population that enhanced their competence for transmission. Demographic dynamics are known to result in substantial temporal variation in the age profile of mosquito populations[35], which has been shown to be a key determinant of dengue epidemic size in some settings[36]. Second, it is also possible that the multiple DENV genotypes in circulation in 2014[37,38] could have been more transmissible than those that had been in circulation previously. One recent study combining laboratory experiments with simulation modeling showed that differences in the extrinsic incubation periods of different genotypes of a single serotype of DENV can vary to such a degree that dengue outbreak sizes could vary by as much as 30%[39]. Third, it is possible that a larger number of unobserved imported infections in 2014 could have given rise to locally acquired cases that were incorrectly attributed to other cases in our analysis. This could be a result of high levels of DENV transmission in other prefectures in Guangdong province in 2014[29] or differences in the proportion of imported infections that were clinically apparent, which could vary from year to year depending on the immunological profile of people who reside elsewhere and visit Guangzhou.

Although it is somewhat unsatisfying that our analysis could not pinpoint the cause of the 2014 epidemic more specifically, clearly defining the roles that known factors played and ruling them out as primary drivers of the 2014 epidemic is also of great value. Our results did show that temperature played a role in delimiting the transmission season, that mosquito density influenced the timing and extent of transmission within a season, and that importation regulated the potential for local transmission in a given season. Our modeling approach was unique in allowing us to isolate each of those effects by building on prior knowledge about them in such a way that we captured their differential influence at different lags and captured the extent to which imported dengue cases translated into locally acquired cases. Had we fitted our model solely to data from 2013 to 2014, as others have[21,23–25], we likely would have misestimated the contributions of these factors to local transmission and would not have been able to detect the anomalous local conditions in 2014 that appear to have driven the large epidemic that year.

Our model incorporates a number of innovations that were essential for obtaining our results, including the ability to accommodate daily incidence data, to adapt the timescale of transmission to the pathogen's generation interval, to estimate multiple lagged effects in a flexible manner, and to isolate the timing of residual variation in transmission, all of which may prove useful to time series analyses of climate-sensitive pathogens[4,40]. At the same time, there are important limitations of our approach. First, even though it is well known that many

DENV infections are inapparent[27], we worked under the assumption that cases detected through passive surveillance were representative of the true incidence of infection. Combining data augmentation methods[41] with hypotheses about ways in which reporting rates might vary through time could offer one way to relax this assumption. Unobserved DENV importation by people[42], or potentially even mosquitoes, could explain some of the residual variation captured by $\beta_0(t)$. Second, we assumed that the population was immunologically naïve and remained so over time. The limited data available pertaining to this question suggest that DENV immunity is indeed low (2.43%, range: 0.28–5.42%)[28], meaning that impacts of immunity on transmission should be negligible. These effects could be stronger at finer spatial scales, however[43]. Third, although the sensitivity analyses that we performed did not indicate a compelling need to incorporate data on local variables other than temperature and mosquito density, there are biological reasons why additional variables, such as precipitation[11] and humidity[44], could be important. Future work in this setting or elsewhere could potentially explain more inter-annual variation in dengue incidence if better ways to leverage additional, biologically appropriate covariates could be devised.

Our finding that epidemic size in any given year depends on a complex interaction between importation and local conditions suggests that public health authorities should not focus on only one of these factors at the exclusion of others. As some studies have done[21–23], it is tempting to attribute the increase in local dengue incidence in Guangzhou to the concurrent increase in imported dengue. Our results suggest that doing so belies the important role that local conditions play in limiting or enhancing transmission in any given year. What an overly simplistic view risks is allowing for another epidemic like the one in 2006, which our results suggest was driven by favorable local conditions despite relatively low importation. Moreover, understanding and reducing the favorability of local conditions for transmission may also mean the difference between years like 2014 and 2015, with importation high in both years but local transmission much lower in 2015. Given the global expansion of DENV and other viruses transmitted by *Aedes* mosquitoes, improved understanding of the interactions among multiple drivers in settings with potential for seasonal DENV transmission—including portions of Australia, the United States, and the Mediterranean—will be essential for reducing the risk of large epidemics such as the one observed in Guangzhou in 2014.

## Methods

**Data**. Data on locally acquired and imported dengue cases from 2005 to 2015 were obtained from the Health Department of Guangdong Province (http://www.gdwst.gov.cn). We considered data from Guangzhou, the capital and most populous city of the Guangdong province, located in southern China with a humid subtropical climate (Supplementary Fig. 9). As a statutorily notifiable infectious disease in China since 1989, dengue is diagnosed according to national surveillance protocol with standardized case definitions described in detail elsewhere[17]. In short, probable and confirmed dengue cases were diagnosed and reported by local physicians according to an individual's epidemiological exposure, clinical manifestations, or confirmed laboratory results. An imported case was defined as one for which the patient had traveled abroad to a dengue-endemic country within 15 days of the onset of illness. In some cases, importation was defined based on laboratory results showing that the infecting DENV had a high sequence similarity in the preM/E region compared with viruses isolated from the putative source region where the patient had traveled. Among all 217 dengue cases imported from other countries, the suspected country of origin was recorded for 204 (94.5%) cases: 76.1% came from Southeast Asia, 13.2% from South Asia, and 4.9% from Africa, with Thailand (22.1%), Malaysia (15.2%), Indonesia (9.3%), the Philippines (7.4%), and Cambodia (7.4%) being the top five countries of origin. In the absence of meeting the criteria for an imported case, a dengue case was considered locally acquired. This determination was made by local public health institutes. All the data used in this study were anonymized; the identity of any individual case cannot be uncovered.

Information about DENV serotype was known for some cases, but not at sufficient resolution to be taken into account in our analysis. The proportion of

locally acquired cases for which DENV serotype was identified ranged from 11.7% (188/1603 reported cases across Guangdong province) during the relatively low-transmission period from 2005 to 2011[28] to 0.8% (345/45,225 reported cases across Guangdong province) during the large epidemic in 2014[29,32]. As summarized elsewhere[28,29], DENV-1 appeared to be the dominant serotype in 2005–2011, with reports of the other three serotypes during 2012–2015. Within a given serotype, multiple genotypes were observed across all years and in 2014 in particular[37,38].

We utilized indices of both adult mosquito density and larval density, which are available from the Guangzhou Center for Disease Control and Prevention (http://www.gzcdc.org.cn). Adult *Ae. albopictus* mosquitoes were sampled by light traps with mosquito ovitrap index (MOI). The MOI was defined as the number of positive ovitraps for adult and larval *Ae. albopictus* per 100 retrieved traps[45]. Breteau index (BI), which measures the density of *Ae. albopictus* mosquito larvae, was the number of positive containers per 100 houses inspected[45,46]. Both indices were measured monthly and comprised the information on mosquito density sampled in residential households (>50 households sampled per month), parks, construction sites, and other urban areas. Data on daily average temperature were obtained from the China Meteorological Data Sharing Service System (http://data.cma.cn).

**Model description.** A general framework that can be used to model the relationship between cases from one generation to the next is the TSIR model[26]. Under an adaptation of that model to realistically account for time lags associated with vector-borne pathogen transmission[47], $I_t$ is defined as the number of new local cases at time $t$ and $I'_t$ is the effective number of cases, both local and imported, that could have generated a local case at time $t$. This effective number of cases in the previous generation is defined as

$$I'_t = \sum_{\tau=1}^{49} \omega_\tau (I_{t-\tau} + \iota_{t-\tau}), \quad (1)$$

where $\omega_\tau$ is the probability that the serial interval is $\tau$ days[47]. We informed the values of $\omega_\tau$ that describe the length of the serial interval using a probability density function derived by Siraj et al.[48] based on first-principles assumptions about DENV transmission. This formulation takes into account lags associated with DENV incubation in humans (intrinsic incubation period), DENV incubation in mosquitoes (extrinsic incubation period), and mosquito longevity, resulting in a probabilistic summary of the time that elapses between one human case and another. The flexibility afforded by Eq. (1) allowed us to fully utilize the daily resolution of case data available for Guangzhou, which distinguished between imported and local cases, $\iota_t$ and $I_t$, respectively.

Consistent with other TSIR models, the relationship between $I'_t$ and $I_t$ was assumed to take the form

$$I_t = \beta(t) \frac{I'_t}{N} S'_t, \quad (2)$$

where $\beta(t)$ is the transmission coefficient on day $t$, $N$ is population size, and $S'_t$ is the number of susceptible people who could potentially become infected and present on day $t$. Due to the low incidence of dengue in Guangzhou on a per population basis (40,108 cases detected by surveillance during 2005–2015 in a city of 14 million), the number of susceptible people at any given time changes very little and remains very close to the overall population size. Therefore, we assumed that $S'_t \approx N$, meaning that these terms canceled out in Eq. (2). Also because of such low incidence, including many days with zero incidence, accounting for the role of stochasticity in transmission was essential. Eq. (2) has a clear and direct stochastic analogue in

$$I_t \sim \text{negative binomial} (\beta(t)I'_t, I'_t), \quad (3)$$

where $\beta(t)I'_t$ is the mean parameter and $I'_t$ is the clumping parameter of the negative binomial distribution[49,50].

We assumed that the potential for local transmission at time $t$, represented by $\beta(t)$, was determined by a combination of latent variables representing mosquito abundance at time $t$, $m(t)$, temperature at time $t$, $T_t$, and other factors not accounted for directly by available data, such as mosquito control or preventative measures taken by local residents. We additionally considered the possibility that relative humidity and precipitation may be contributing to transmission patterns, but did not include these in our final model as the overall correlations between simulated local incidence and observed local incidence for these models was poorer than the model that only considered temperature and mosquito abundance (Supplementary Methods 2). Although the role of these factors in driving transmission is commonly assumed by models[51] and consistent with the highly seasonal nature of DENV transmission in Guangzhou[17], it is also clear that these factors may influence transmission considerably in advance of a case occurring. For example, high mosquito densities would be expected to affect transmission 2–3 weeks in advance, rather than instantaneously, to allow mosquitoes sufficient time to become infected, incubate the virus, and transmit it[48].

To afford the model sufficient flexibility to account for such lagged effects, we allowed $\beta(t)$ to depend on weighted sums of daily effects of $m(t-\tau)$ and $T_{t-\tau}$ for $\tau \in \{1, …, 49\}$, which spanned the full range of serial intervals that we assumed

were possible based on the serial interval formulation by Siraj et al.[48]. Because the effects of $m(t-\tau)$ and $T_{t-\tau}$ could differ for different values of $\tau$ in complex ways, we defined flexible bivariate basis functions $s_m(m(t-\tau), \tau))$ and $s_T(T_{t-\tau}, \tau)$ with cubic B-splines to capture the contribution of daily conditions on day $t-\tau$ to $\beta(t)$ using the fda package in R[52]. Although not represented explicitly, the $s_T(T_{t-\tau}, \tau)$ function is sufficiently flexible to account for the combined effects of the temperature-sensitive virus and vector traits, such as the extrinsic incubation period, mosquito mortality, and mosquito bites. Whereas other models represent those factors explicitly based on mechanistic assumptions, we model their combined influence in a more statistical fashion. Mathematically, each of $s_m(m(t-\tau), \tau)$ and $s_T(T_{t-\tau}, \tau)$ was defined by nine parameters associated with a $3 \times 3$ matrix that defined the height of each component of the bivariate spline ranging 1–49 days for $\tau$, 4–36 °C for $T$, and 0–5 for $m$, with the units of the latter corresponding to the scale of the mosquito oviposition index. That particular choice of units was not of consequence to the model, however, because a different choice would simply result in different values of parameters in $s_m(m(t-\tau), \tau)$ but yield the same values of $\beta(t)$.

These lagged daily effects combined to define

$$\beta(t) = e^{\sum_{\tau=1}^{49} s_T(T_{t-\tau}, \tau)} e^{\sum_{\tau=1}^{49} s_m(m(t-\tau), \tau)} e^{\beta_0(t)}, \quad (4)$$

where $\beta_0(t)$ is a univariate cubic B-spline function that defines the time-varying contribution of local factors other than temperature and mosquito abundance to $\beta(t)$. We interpret $\beta_0(t)$ as corresponding to local factors due to the fact that it modulates local transmission in our model—i.e., how many locally acquired cases result from each case in the preceding generation. Mathematically, we specified $\beta_0(t)$ as a univariate spline with three evenly spaced knots per year across the 11-year time period, requiring a total of 33 parameters. We also represented the latent mosquito density variable $m(t)$ using a univariate cubic B-spline function with three knots per year for the 11-year time period. This variable allowed us to reconcile differences between the MOI and BI mosquito indices and to obtain daily values for mosquito abundance based on monthly indices.

**Model fitting.** We used a two-step process to estimate the posterior probability distribution of model parameters. First, we fitted the entomological model (i.e., $m(t)$) using maximum likelihood. Second, we fitted the epidemiological model using a Sequential Monte Carlo (SMC) algorithm in the BayesianTools R library[53].

For the entomological likelihood, the probability of the full mosquito index time series, $\overrightarrow{\text{MOI}}$ and $\overrightarrow{\text{BI}}$, depends on the 33 parameters that define $m(t)$ (referred to collectively as $\overrightarrow{\theta}_m$) and three parameters, $\mu_{BI}$, $\sigma_{BI}$, and $\sigma_{MOI}$, that define an observation model relating $m(t)$ to the data. Under this model, the probabilities of these data are

$$\Pr\left(\overrightarrow{\text{MOI}} \middle| \overrightarrow{\theta}_m, \sigma_{MOI}\right) = \prod_t \phi(\text{MOI}_t | \overline{m}(t), \sigma_{MOI}) \quad (5)$$

and

$$\Pr\left(\overrightarrow{\text{BI}} \middle| \overrightarrow{\theta}_m, \mu_{BI}, \sigma_{BI}\right) = \prod_t \phi(\text{BI}_t | \mu_{BI}\overline{m}(t), \sigma_{BI}), \quad (6)$$

where $\phi(x|\mu, \sigma)$ denotes a normal probability density with parameters $\mu$ and $\sigma$ evaluated at $x$ and $\overline{m}(t)$ denotes the monthly average of $m(t)$. Together, Eqs. (5) and (6) specify

$$\mathcal{L}\left(\overrightarrow{\theta}_m, \sigma_{MOI}, \mu_{BI}, \sigma_{BI} \middle| \overrightarrow{\text{MOI}}, \overrightarrow{\text{BI}}\right) =$$
$$\Pr\left(\overrightarrow{\text{MOI}} \middle| \overrightarrow{\theta}_{m,\sigma_{MOI}}\right) \Pr\left(\overrightarrow{\text{BI}} \middle| \overrightarrow{\theta}_{m,\mu_{BI},\sigma_{BI}}\right), \quad (7)$$

which is the overall likelihood of the entomological model and its parameters.

We obtained maximum-likelihood estimates of $\overrightarrow{\theta}_m$, $\sigma_{MOI}$, $\mu_{BI}$, and $\sigma_{BI}$ by maximizing the log of Eq. (7) using the Nelder-Mead optimization algorithm under default settings in the optim function in R[54]. To safeguard against obtaining an estimate that represented a local rather than global optimum, we repeated this optimization procedure 1000 times under different initial conditions. The initial conditions for each of these runs came from separate draws from a posterior distribution obtained through SMC estimation using the BayesianTools R library[53]. Of the 1000 estimates of $\overrightarrow{\theta}_m$, $\sigma_{MOI}$, $\mu_{BI}$, and $\sigma_{BI}$ that this yielded, we chose the one with the highest log likelihood to derive our maximum-likelihood estimate of $m(t)$ for use in the epidemiological model (Supplementary Fig. 8).

For the epidemiological likelihood, the probability of the local incidence data, $\overrightarrow{I}$, depends on a total of 51 parameters in addition to $m(t)$ that define $\beta(t)$, with nine for $\overrightarrow{\theta}_{s_m}$, nine for $\overrightarrow{\theta}_{s_T}$, and three for each of the eleven years spanned by $\overrightarrow{\theta}_0$. Although the transmission model (Eq. 3) is stochastic, it does not readily lend itself to calculation of the probability of $\overrightarrow{I}$ as a function of these parameters. Consequently, we used a simulation-based approach to approximate the probability of each daily value of $I_t$ under a given value of the 51 model parameters. To do so, we performed 100 simulations of the entire time series of local incidence

across 2005–2015, with each simulation driven by data on imported cases feeding into Eq. (3) for a given $\beta(t)$. As new local cases were generated in these simulations of local transmission, those new local cases fed back into generating subsequent local cases, again following Eq. (3). Using these simulations, we approximated a probability of the local incidence data by treating the number of local cases on a given day as a beta-binomial random variable. This assumes that all residents of Guangzhou are subject to a probability of being infected and detected by surveillance as a locally acquired dengue case on each day. Uncertainty in that probability was assumed to follow a beta distribution with parameters informed by the ensemble of simulated incidence. Specifically, parameters of this beta distribution followed Bayesian conjugate distributional relationships as $\alpha_t = 1 + \sum_{i=1}^{100} I_{t,i}$ and $\beta_t = N - \sum_{i=1}^{100} I_{t,i} + 1$[55], where $N = 14{,}040{,}000$. This effectively treats simulated incidence values as prior observations that inform a posterior estimate of the daily probability of infection with DENV as a beta random variable. From there, the probability of a given incidence $I_t$ on day $t$—and the contribution of that data point to the likelihood of the model parameters used to simulate those incidence patterns—is calculated according to a beta-binomial distribution, which treats each individual's infection outcome as resulting from a Bernoulli trial with probability informed by the beta distribution. In summary, 100 values of $I_{t,i}$ simulated for each day in 2005–2015 using a single $\beta_t$ specified

$$\mathcal{L}\left(\vec{\theta}_{sm},\ \vec{\theta}_{sT},\ \vec{\theta}_0 \middle| \vec{I},\ \vec{T},\ m(t)\right) = \prod_t \text{beta binomial}\left(I_t | \alpha_t, \beta_t, N\right) \quad (8)$$

as the overall likelihood of the epidemiological model parameters.

Given that numerous studies have investigated relationships between temperature, mosquito density, and DENV transmission, we sought to leverage that information by specifying prior distributions for epidemiological model parameters. Doing so still permits the data to influence parameter estimates in the posterior via the likelihood, but it does so in such a way that parameter values in the posterior are penalized somewhat if they deviate strongly from prior understanding of which parameter values are plausible based on previous work. For $\vec{\theta}_{s_m}$ and $\vec{\theta}_{s_T}$, we used relationships between $T$, $m$, and $R_0$ (which is similar to our transmission coefficient $\beta$[47]) recently described by Siraj et al.[48]. In doing so, we assumed that relationships among these variables were identical at all lags $\tau$, given a lack of specific prior understanding of how these relationships vary at different lags. Given that the scales of $m$ and that of Siraj et al.[48] are not directly comparable, we parameterized the prior distribution around values of $m$ with relevance to the time series of $m(t)$ in Guangzhou. That is, at the temperature optimum of 33.3 °C estimated by Siraj et al.[48], we set our prior for $\beta$ such that $\beta = 0$ when $m = 0$ and $\beta = 2.5$ when $m = 3$. The latter value of $m$ is just above the maximum value estimated for Guangzhou, and the corresponding value of $\beta$ is equal to the median seasonal estimate of daily $R_0$ in Iquitos, Peru, a dengue-endemic setting with empirical estimates of seasonal $R_0$[56] that Guangzhou should be unlikely to exceed. At the same time, posterior estimates of the parameters did have the flexibility to yield values of $\beta$ in excess of 2.5 should the data support such values via the likelihood. Consistent with standard theory for mosquito-borne disease epidemiology[13], values of the prior at other temperatures were obtained by reducing the value of $\beta$ linearly in proportion to $m$ and by the proportion of $R_0$ from Siraj et al.[48] for other temperatures relative to its value at 33.3 °C. Using 1000 Monte Carlo samples of the relationship between $T$, $m$, and $R_0$ from Siraj et al.[48], we obtained 1000 estimates of $\vec{\theta}_{s_m}$ and $\vec{\theta}_{s_T}$ by using the optim function in R to minimize the sum of squared differences between $R_0$ values from Siraj et al.[48] and corresponding values of $\beta$ defined by $\vec{\theta}_{s_m}$ and $\vec{\theta}_{s_T}$ and with $\beta_0 = 0$. A multivariate normal distribution fitted to those 1000 estimates of $\vec{\theta}_{s_m}$ and $\vec{\theta}_{s_T}$ represented our prior distribution of those parameters. Separately, we defined the prior distribution of each parameter in $\vec{\theta}_0$ as normally distributed with mean 0 and standard deviation 5, given our expectation that residual variation in $\beta(t)$ not attributable to temperature or mosquito density should be minimal, on average.

We obtained an estimate of the posterior distribution of epidemiological parameters using an SMC algorithm implemented in the BayesianTools R library[57]. To assess convergence, we performed three independent runs of the SMC algorithm set to ten iterations of 10,000 samples each (Supplementary Figs. 6, 7). We then calculated the Gelman-Rubin convergence diagnostic statistic across the three independent runs, along with the multivariate potential scale reduction factor (Supplementary Table 2)[58].

**Simulation experiments**. To verify that the behavior of the transmission model was consistent with the data to which it was fitted, we simulated an ensemble of 2000 realizations of daily local incidence using parameter values drawn from the estimated posterior distribution. These simulations were performed for all of 2005–2015 in the same manner in which the likelihood was approximated; i.e., driven by imported case data and with local transmission following Eqs. (1–4). We compared simulated and empirical local incidence patterns in two ways. First, we computed Pearson's correlation coefficient between daily local incidence data and median values from the simulation ensemble. Second, we compared simulated and empirical patterns on an annual basis in terms of four features of local incidence patterns: annual incidence, peak weekly incidence, total number of weeks with

non-zero local incidence, and number of weeks between the first and last local case. Consistency between simulated and empirical values of these quantities was assessed using Bayesian p-values, with values >0.025 and <0.975 indicating consistency between empirical values and the model-derived ensemble[55].

To partition inter-annual variation in local incidence into portions attributable to inter-annual variation in local conditions or importation patterns, we performed a simulation experiment with a two-way factorial design. In this experiment, we grouped temperature, mosquito density, and residual variation in local conditions together as one set of predictor variables and importation patterns as the other. Each year from 2005 to 2015 was considered as a factor for each set of predictors. An ensemble of 1000 simulations was generated for each of the 122 combinations of 11 years of each of the two sets of predictors. For example, with temperatures, mosquito densities, and $\beta_0(t)$ values from 2005, 1000 simulations were performed with imported cases from each of 2005–2015, and likewise for temperatures, mosquito densities, and $\beta_0(t)$ values from 2006 to 2015. We summed annual local incidence for each of these 122,000 simulations and performed a two-way analysis of variance, resulting in estimates of the variation (defined in terms of sum of squared error, SSQ) in annual incidence attributable to local conditions, to importation, and to a portion unexplained by either predictor set due to the stochastic nature of the simulations. Because the number of 1000 replicates was at our discretion in this simulation experiment, the p-value from this analysis of variance was not meaningful[59].

To quantify the overall portion of variation attributable to each predictor variable, we performed an additional simulation experiment with a four-way factorial design. In this experiment, we interchanged temperature, mosquito density, $\beta_0(t)$ values, and importation patterns from different years, again considering each year as a factor for each predictor variable. An ensemble of 1000 simulations was generated for each of the 14,641 combinations of 11 years of all four predictors. Similar to the two-way factorial experiment, we summed annual local incidence for each of these simulations and performed a four-way analysis of variance. This resulted in direct estimates of the variation in annual incidence attributable to temperature, mosquito density, $\beta_0(t)$ values, importation patterns, and to a portion attributable to stochasticity.

**Reporting summary**. Further information on experimental design is available in the Nature Research Reporting Summary linked to this article.

## Data availability
Data, code, and description of how to replicate the analyses are available on GitHub at https://github.com/roidtman/NatComm_dengue_China.

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

## Acknowledgements

R.J.O. acknowledges support from an Arthur J. Schmitt Leadership Fellowship in Science and Engineering and an Eck Institute for Global Health Fellowship. S.L. acknowledges supports from the National Natural Science Fund (No. 81773498) and the National Science and Technology Major Project of China (2016ZX10004222-009). J.Y. acknowledges support from the National Science and Technology Major Project of China (2018ZX10201001). A.J.T. acknowledges support from the Bill & Melinda Gates Foundation (OPP1106427, 1032350, OPP1134076, OPP1094793), the Clinton Health Access Initiative, the UK Department for International Development (DFID), and the Wellcome Trust (106866/Z/15/Z, 204613/Z/16/Z). T.A.P. acknowledges support from the National Science Foundation (DEB 1641130) and the Defense Advanced Research Projects Agency (D16AP00114). H.Y. acknowledges support from the National Natural Science Fund for Distinguished Young Scholars of China (No. 81525023), Program of Shanghai Academic/Technology Research Leader (No. 18XD1400300), the United States National Institutes of Health (Comprehensive International Program for Research on AIDS grant U19 AI51915).

## Author contributions

S.L., Z.H., T.A.P. and H.Y. conceived of the study. R.J.O., S.L., Z.H. and J.Y. curated the data. R.J.O., A.S.S. and T.A.P. performed the formal analysis. Funding acquisition was by T.A.P. and H.Y. Investigation was by R.J.O. Methodology was developed by R.J.O., R.C.R. and T.A.P. Project administration was by R.J.O. and S.L. H.Y. supplied resources. R.J.O. worked with the software. A.J.T., T.A.P. and H.Y. supervised the project. R.J.O. and T.A.P. validated the model. R.J.O., T.A.P. and A.S.S. produced visualizations. R.J.O., S.L. and T.A.P. wrote the original draft of the manuscript. All authors contributed to the final version of the manuscript.

## Additional information

**Competing interests:** The authors declare no competing interests.

