## [Peer Review File · Nature Communications]

Reviewers' comments:

Reviewer #1 (Remarks to the Author):

In the proposed manuscript, Oidtman et al. use a semi-mechanistic time series model to explore the potential factors contributing to inter-annual variation in dengue virus (DENV) transmission in Guangzhou, China from 2005-2015. The authors claim that a combination of favorable factors, including temperature, mosquito density, imported cases and other unexplained anomalies facilitated the development of a large dengue epidemic in 2014.

The manuscript is well written and the modelling approach seems mathematically sound. The results emphasise the importance of identifying the potential mechanisms, not explicitly represented in the model, which contributed to the high dengue incidence in 2014. I have the following questions about the model inputs, presentation and interpretation of the modelling results:

1. The authors selected daily average temperature from the China Meteorological Data System. However, previous studies (outlined in Table S1) concluded that precipitation was also an important factor for predicting dengue outbreaks in Guangzhou. Fig 1c shows very little interannual variability in temperature. Have the authors explored interannual variability in other meteorological variables?

2. The authors include the term $\beta_0(t)$ in the model to account for residual variation caused by unexplained factors. They group this term with local conditions (temperature and mosquito density). However, this unknown variation could also arise from external factors or within country importation. In the presentation of results, it would be useful to separate this term from the 'local' conditions group (e.g. show local, imported and unexplained variation components). For example, in Table 1, separate $\beta_0(t)$ from M+T. It was unclear to me what SSQ_res represents. Is this different to SSQ_ $\beta_0(t)$? It would be helpful if each component in rows 1 and 2 summed to 100%. The authors state that the overall variation explained when including all years was 51.2%. How much variation was explained by the model excluding 2014? I suggest adjusting the discussion to remove reference to 'local' when referring to the unexplained anomalous conditions in 2014.

3. How does this complex model compare to a simple model (e.g. a regression model for daily dengue incidence, using entomological data, daily temperature and a time-varying random effect to capture residual variation)?

Minor points

-To what extent does the inclusion of $\beta_0(t)$ contribute to the ranking of years simulated from the fitted model? How does the ranking change if this term is excluded from the model?

-Include a description and map of the study area in the methods.

-Can the authors provide a breakdown or summary of imported case origin?

Line 44-45: Rewrite sentence "However, different models have identified different drivers, leading to inconsistent conclusions."

Line 76: Specify the two types of entomological data (e.g. MOI, BI).

Line 93: Quantify intermediate, shorter and longer lags.

Line 124: Define the shorter seasonal time window and provide an example.

Line 196: "Because the number of replicates was at our discretion in this simulation experiment, the p-value from this analysis of variance was not meaningful". Remove or move this sentence to methods or supplementary.

Line 250: "anomalous local conditions" (change local to unexplained).

Line 254: "unknown local conditions" (remove local).

Line 368: Would a zero-inflated negative binomial distribution help in this case?

Line 369: Consider changing clumping to overdispersion.

Line 374: latent variable[s]

Line 443: "As local incidence occurred in these simulations". Do the authors mean the model estimated cases to occur in the time period 2005-2015?

Fig 1. Include a legend in Figure 1C and describe symbols and curve in the caption.

Fig 2. State 1-49 days.

Fig 3. Include labels A, B, C and D before each item.

Fig 5 and 7. Consider using a sequential rather than diverging colour bar for the ranks (e.g. dark to pale red rather than dark red to dark blue).

Fig. 6. Consider using a log scale for consistency between figures.

Fig. 7. Do the figures show results using a specific year for local conditions (middle) and imported cases (bottom)? If so, specific the year in the caption.

Table S1: Row 5, dominant factors of dengue fever transmission.

Table S2: Define SMC.

Reviewer #2 (Remarks to the Author):

Overall this paper provides some unique insights on the epidemic that occurred in Guangzhou, China. The results are inconclusive but they point towards a need for closer examination of varying local factors being the dominant factor in the epidemic. This is somewhat interesting and provides a jumping off point for further investigations. This is a solid contribution to the literature, but has potential for significant improvement and increased impact if more consideration of key components of the transmission dynamics, temperature driven EIP, humidity, and serotype (only discussed minimally) were incorporated more into the analyses and discussion. Authors should be careful to clearly articulate the overall findings of studies that have previously examined this dataset as indicated below and how they differ as this dataset has been used relatively extensively.

The analyses seem appropriate, but I am admittedly not a biostatistician. If this paper is combined with other referenced papers, it should be possible to recreate the analyses.

Introduction

Line 15 – It is important to note that while some regions are becoming more suitable, others are becoming less suitable for specific pathogens. These relationships are complex and it may be better to discuss the role of climate change in terms of altering current transmission patterns instead of solely as increasing.

Line 16 – authors note that El Nino is a clear example, but they do not reference the pathway or the direction. Clarify.

Line 28 – “pathogen importation” that occurs at times concurrent with weather conditions suitable for transmission – should be noted. If infected individuals arrive during the time period with very low mosquito density, for example, an imported case may have minimal influence.

Line 30 – The authors move from a very general discussion of climate / climate change and population introduction of pathogens to a specific history of an epidemic of dengue in Guangzhou, China. It would be beneficial to have a third paragraph that bridges this more general discussion with some specifics related to dengue and climate change. While some of the references are on dengue in the first two paragraphs, there are an equivalent number on cholera, malaria, etc.

Line 35-37 – Case numbers are fine but not comparable across geographic areas. Since this paragraph is setting up the differential transmission in SE Asia vs. Guangzhou, I recommend including the cases per 100,000 population.

Line 37 and Line 43 – The case numbers from 2014 are variably presented as 47,056 and 37,445. Confirm.

Lines 46, 48 – If referring to specific temperature/ humidity/ precip. etc. for 2014, this should be referred to as weather patterns, not climate which typically refers to periods of 20 years or more.

Lines 44-53 – The review of the articles that have used this dataset prior, indicates an either – or type of result, with some finding that importation is the leading factor and other that “climate” (weather) is the leading factor, however, Cheng et. al. 2016 and Cheng et. al. 2017 both appear to indicate that it is both related to anomalous weather and timing of imported cases. I would suggest revising this paragraph to better reflect the findings from these initial analyses of the data.

Methods:

Line 322 – Dengue case definitions are deferred to another paper, but including a one sentence synopsis is important. Are these suspected or lab confirmed?

Line 391 – The rationale for the 1-49 day range is not clear. How was the full range of serial intervals determined?

Line 448-450 – More details on the choice of specific beta distribution parameters for probability of being infected would be useful.

Line 468 – The optimum temperature is indicated as 33.3oC – which seems quite high given results from temperature-survival experiments for *Ae. albopictus*. It is anticipated that this is not just accounting for the optimum mosquito survival/ density but takes into account the EIP which shortens considerably as temperature increases.

Further, it does not appear that the relationship between the EIP and temperature is considered explicitly in the modeling framework. This would alter how the three components of the model interact.

Results:

Lines 90-95: Authors describe differences in the contribution of short, intermediate, and long lag

times. It would be useful to define these time periods in terms of days. Suggest color gradation in Fig. 2B to differentiate these more clearly.

Lines 120-124: Authors discuss the relative influence of temperature and mosquito density. In the discussion the authors make mention of the importance of temperature on mosquito longevity as a potential explanation of relationships found (lines 265-266). Humidity also influences longevity and could be another potential variable included if there is significant differences (Schmidt et. al. Parasites and Vectors 2018). Another very important parameter that is not discussed is the significant relationship between temperature and the extrinsic incubation period of the virus. This is a critical parameter in determining transmission potential and should receive some attention within the manuscript both in the modeling approach and the interpretation of findings.

Line 171: What is meant by "local conditions", is this just the mosquito and temperature components of the model?

Line 177: Why were 2008 importation conditions used as the comparator across the years? Was it highest? Lowest?

Discussion:

Authors discuss very briefly the fact that the serogroup that year may have been more infectious and/or pathogenic leading to a greater portion of cases being reported. It would be helpful if any information on serogroup in the region over time could be added and discussed. It is fairly well established that DENV 2 is related to more severe illness. If this was the serotype circulating it would support their argument.

Further serotype is a critical component of interannual dengue transmission variability and even if susceptibility is high for all serogroups due to lower levels of historical transmission, sending communities (i.e. those that are the sources of imported transmission) that have high levels of transmission each year would potentially result in greater numbers of imported cases to seed the transmission Guangzhou in multiple geographic areas. The proportion of infected individuals that are traveling into the area that get captured in the surveillance system is likely quite low, though the proportion would not necessarily be expected to vary annually. It is also possible that even if the number of imported cases is not driving the epidemic, spatial patterns of imported cases could further explain some of the high transmission in 2014. It is likely that only a small portion of imported cases are detected.

Even if serogroup information in the region is spotty, including some evidence about circulating serotypes would be helpful. Since line 325 indicates that there is sequencing conducted on viral isolates, it would seem this data is available somewhere to enrich the discussion.

Another potentially interesting factor is the role of imported mosquitoes. Clearly this is not possible to include in an analysis but if transmission across the region was high in 2014, there is a possibility of introduction of virus through infected mosquitoes through importation of goods. Inclusion of information on the overall transmission dynamics regionally would enhance the discussion.

Reviewers' comments:

Reviewer #1 (Remarks to the Author):

In the proposed manuscript, Oidtman et al. use a semi-mechanistic time series model to explore the potential factors contributing to inter-annual variation in dengue virus (DENV) transmission in Guangzhou, China from 2005-2015. The authors claim that a combination of favorable factors, including temperature, mosquito density, imported cases and other unexplained anomalies facilitated the development of a large dengue epidemic in 2014.

The manuscript is well written and the modelling approach seems mathematically sound. The results emphasise the importance of identifying the potential mechanisms, not explicitly represented in the model, which contributed to the high dengue incidence in 2014. I have the following questions about the model inputs, presentation and interpretation of the modelling results:

Comment 1.1

1. The authors selected daily average temperature from the China Meteorological Data System. However, previous studies (outlined in Table S1) concluded that precipitation was also an important factor for predicting dengue outbreaks in Guangzhou. Fig 1c shows very little interannual variability in temperature. Have the authors explored interannual variability in other meteorological variables?

Response 1.1

In response to the reviewer's comment, we added a new Supplemental Text 2 in which we explored the inclusion of additional meteorological variables. There, we explored four alternative model formulations, three of which included the addition of rainfall, relative humidity, or rainfall and relative humidity. Simulations from the model used in our primary analysis—which only included temperature, mosquito density, and a time-varying residual term—had the highest Pearson's correlation coefficient with the empirical data (Table ST2.1), best reproduced the rankings of years by incidence in our data set (compare Figs. ST2.10-ST2.13 with Fig. 5), and had the most years with Bayesian p-values between 0.025 and 0.975 for the four epidemic features of interest (Table ST2.1). Because our original model performed best by all of these measures, we chose to retain it for our primary analysis described in the main text. This, as well as reconciliation of this conclusion with the known biological significance of precipitation and humidity, is discussed in Supplemental Text 2 on lines 93-99. In the main text, we referenced alternative models in Supplemental Text 2 on lines 119, 177, and 475-476.

Comment 1.2

2. The authors include the term $\beta_0(t)$ in the model to account for residual variation caused by unexplained factors. They group this term with local conditions (temperature and mosquito density). However, this unknown variation could also arise from external factors or within country importation. In the presentation of results, it would be useful to separate this term from the 'local' conditions group (e.g. show local, imported and unexplained variation components). For example, in Table 1, separate $\beta_0(t)$ from M+T. It was unclear to me what SSQ_res represents. Is this different to SSQ_beta_0(t)? It would be helpful if each component in rows 1 and 2 summed to 100%. The authors state that the overall variation explained when including all years was 51.2%. How much variation was explained by the model excluding 2014? I suggest adjusting the discussion to remove reference to 'local' when referring to the unexplained anomalous conditions in 2014.

Response 1.2

The reviewer makes a fair point that there are relevant sources of uncertainty that could derive from a range of internal and external factors. In light of the structure of our model, however, we feel that it is appropriate to refer to unspecified factors accounted for by $\beta_0(t)$ as local conditions, because they enter our model through their influence on the extent to which cases in one generation lead to cases in the next generation through a transmission process that occurs locally. Additional importation simply would not enter into our model in this way. Accounting for unobserved importations, whether they be of domestic or foreign origin, would require adding an additional term to the model to augment the imported cases currently used to seed local transmission. Although doing so using data augmentation techniques (e.g., Reference 41) would be very interesting, we feel that doing so would be a significant undertaking and would be more appropriate for future work. Nonetheless, to incorporate the reviewer's insights on this issue into the manuscript, we have more clearly articulated our reasoning for referring to factors accounted for by $\beta_0(t)$ as local conditions in the Introduction on lines 83-85. We have also acknowledged in the Discussion on lines 361-362 that $\beta_0(t)$ could be capturing some influence from unobserved importations but that investigating this is beyond the capability of our model's current structure. Consistent with our view that it is appropriate to view $\beta_0(t)$ as an aspect of local conditions due to its place in our model's term for local transmission, we refrained from performing the additional factorial simulation experiment suggested by the reviewer to disentangle the contributions of inter-annual variation in M+T and $\beta_0(t)$ on inter-annual variation in local incidence. Ultimately, we feel that all three of these components of our model's term for local transmission should be viewed as varying together across years. We have, however, made adjustments to Tables 1 & 2 consistent with the reviewer's suggestions.

Comment 1.3

3. How does this complex model compare to a simple model (e.g. a regression model for daily dengue incidence, using entomological data, daily temperature and a time-varying random effect to capture residual variation)?

Response 1.3

To address this question, we fitted a simpler model along the lines of what the reviewer suggested and describe this exercise in Supplemental Text 1. To do so, we followed methods from Kraemer et al. (2018), which are still rooted in the same TSIR framework we used here but using a much simpler regression approach based on model predictions of incidence from one generation to the next (~3 weeks). This contrasts with our simulation-based approach, which incorporated not only the model's ability to predict incidence one generation ahead but to recreate the entire time series. Consistent with Kraemer et al., we fitted incidence predictions from one generation to the next using a shape-constrained additive model, which is a type of regression model that allows for flexibility in the shape of relationships between predictor and response variables but likewise allows for the imposition of sensible constraints on the shapes of those relationships where desired. Consistent with our more complex model, we included a concave relationship between temperature and incidence, an increasing relationship between mosquito density and incidence, and a flexible, time-varying term to capture residual variation. These same elements of our more complex model were captured by this simpler model in a single line of R code and are, therefore, consistent with our interpretation of the rationale for the reviewer's comment. Our findings from this exercise indicate that this simple model was not capable of accurately reproducing observed patterns of local incidence (simple model with $\rho = 0.269$, compared with complex model with $\rho = 0.966$). Had we adopted this model for our primary analysis, it would have failed in the sections in the Methods and Results titled "Checking model consistency with data." This is a crucial issue, because without a model capable of recreating these patterns, we would not have been able to use the model to conduct the simulation experiments that were centrally important to addressing our motivating questions about drivers of inter-annual variation in dengue incidence. As a result of this process and similar experiences during the early stages of our work on this project, we concluded that the more complex model we presented was necessary to address our motivating questions. We now mention this and refer to Supplemental Text 1 on lines 173-175 in the main text.

Minor points

Comment 1.4

-To what extent does the inclusion of $\beta_{0(t)}$ contribute to the ranking of years simulated from the fitted model? How does the ranking change if this term is excluded from the model?

Response 1.4

To address this comment, we fitted and validated a model without this term (Supplementary Text 2, Alternative Model 1). When we ranked years according to simulated local annual incidence, we observed that 2014 was not predicted to be the year with the highest simulated local annual incidence (compare Figs. ST2.10-ST2.13 with Fig. 5). This is an important requirement for any model used to address our primary questions about drivers of inter-annual variation in dengue incidence, so we added several figures in Supplemental Text 2 to further describe this model and addressed this point in the main text on lines 118-121 and in Supplemental Text 2 on lines 62-65 and 69-71.

Comment 1.5

-Include a description and map of the study area in the methods.

Response 1.5

We added a basic description of Guangzhou into the Methods on lines 394-396. We added a map of Guangzhou into the supplementary material (Figure S9).

Comment 1.6

-Can the authors provide a breakdown or summary of imported case origin?

Response 1.6

We added a detailed description of the origin of imported cases on lines 405-408.

Comment 1.7

Line 44-45: Rewrite sentence "However, different models have identified different drivers, leading to inconsistent conclusions."

Response 1.7

We revised this sentence on line 59.

Comment 1.8

Line 76: Specify the two types of entomological data (e.g. MOI, BI).

Response 1.8

We specified the two types of entomological data on lines 93-94.

Comment 1.9

Line 93: Quantify intermediate, shorter and longer lags.

Response 1.9

We clarified this on lines 112-113.

Comment 1.10

Line 124: Define the shorter seasonal time window and provide an example.

Response 1.10

We clarified this on lines 146-149.

Comment 1.11

Line 196: "Because the number of replicates was at our discretion in this simulation experiment, the p-value from this analysis of variance was not meaningful". Remove or move this sentence to methods or supplementary.

Response 1.11

We moved this sentence to the Methods on lines 640-641.

Comment 1.12

Line 250: "anomalous local conditions" (change local to unexplained).

Response 1.12

We made this change on line 295.

Comment 1.13

Line 254: "unknown local conditions" (remove local).

Response 1.13

We made this change on line 301.

Comment 1.14

Line 368: Would a zero-inflated negative binomial distribution help in this case?

Response 1.14

We can appreciate the intuition behind the reviewer's suggestion, which would be a very sensible way to deal with this situation in many cases. However, including a zero-inflated term presents some difficulties given how our model works. The stochastic version of the TSIR model (see Xia et al.⁴⁹ for a lucid derivation) was derived in such a way that a negative binomial distribution results as a natural approximation for the birth-death stochastic process of transmission from one generation to another. This formulation has a very specific requirement that the parameters of the negative binomial be applied under this interpretation as $\beta(t)I'_t$ and I'_t , as indicated by our Equation 3. Likewise, predictions about the probability that $I_t = 0$ on a given day t are prescribed by that negative binomial distribution, without zero inflation. Although incorporating zero inflation could possibly help fit the data better, it would distort the interpretation of the model as being reflective of the local transmission process implied by the model's derivation. After taking these issues and the reviewer's suggestion under consideration, we concluded that it was best to not consider a zero-inflated negative binomial distribution in our analysis. Doing so would be perfectly appropriate were our approach purely statistical in nature, but it would not be compatible with the assumptions underlying the mechanistic elements of our model formulation, which is ultimately a hybrid between mechanistic and statistical approaches.

Comment 1.15

Line 369: Consider changing clumping to overdispersion.

Response 1.15

We can understand the reviewer's suggestion to consider changing this to overdispersion parameter given the common use of that term, but we ultimately decided to retain our use of the term clumping parameter given its use in the paper by Xia et al.⁴⁹ that we refer to for derivation of the stochastic TSIR model.

Comment 1.16

Line 374: latent variable[s]

Response 1.16

We made this change on line 469.

Comment 1.17

Line 443: "As local incidence occurred in these simulations". Do the authors mean the model estimated cases to occur in the time period 2005-2015?

Response 1.17

We clarified this point on lines 550-556.

Comment 1.18

Fig 1. Include a legend in Figure 1C and describe symbols and curve in the caption.

Response 1.18

We added a description of the symbols and curve in the caption for Figure 1C.

Comment 1.19

Fig 2. State 1-49 days.

Response 1.19

We made this change in the caption for Figure 2.

Comment 1.20

Fig 3. Include labels A, B, C and D before each item.

Response 1.20

We made this change in Figures 2 and 3.

Comment 1.21

Fig 5 and 7. Consider using a sequential rather than diverging colour bar for the ranks (e.g. dark to pale red rather than dark red to dark blue).

Response 1.21

In response to this comment, we explored the reviewer's suggestion, as well as several other color scheme possibilities. After considering these different options, we ultimately decided to retain the diverging color bar, given our desire to highlight both extremes. Because the central theme of our paper involves accounting for the full range of inter-annual variation in local dengue incidence, we feel that the diverging color bar most clearly emphasizes years with consistently high simulated local incidence (dark red) versus years with consistently low simulated local incidence (dark blue).

Comment 1.22

Fig. 6. Consider using a log scale for consistency between figures.

Response 1.22

We made this change to Figure 6.

Comment 1.23

Fig. 7. Do the figures show results using a specific year for local conditions (middle) and imported cases (bottom)? If so, specific the year in the caption.

Response 1.23

We updated the text in the caption for Figure 7 to clarify this.

Comment 1.24

Table S1: Row 5, dominant factors of dengue fever transmission.

Response 1.24

We made this change in Table S1.

Comment 1.25

Table S2: Define SMC.

Response 1.25

We clarified in the caption for Table S2 that SMC refers to Sequential Monte Carlo.

Reviewer #2 (Remarks to the Author):

Overall this paper provides some unique insights on the epidemic that occurred in Guangzhou, China. The results are inconclusive but they point towards a need for closer examination of varying local factors being the dominant factor in the epidemic. This is somewhat interesting and provides a jumping off point for further investigations. This is a solid contribution to the literature, but has potential for significant improvement and increased impact if more consideration of key components of the transmission dynamics, temperature driven EIP, humidity, and serotype (only discussed minimally) were incorporated more into the analyses and discussion. Authors should be careful to clearly articulate the overall findings of studies that have previously examined this dataset as indicated below and how they differ as this dataset has been used relatively extensively.

The analyses seem appropriate, but I am admittedly not a biostatistician. If this paper is combined with other referenced papers, it should be possible to recreate the analyses.

Introduction

Comment 2.1

Line 15 – It is important to note that while some regions are becoming more suitable, others are becoming less suitable for specific pathogens. These relationships are complex and it may be better to discuss the role of climate change in terms of altering current transmission patterns instead of solely as increasing.

Response 2.1

We made this change on lines 14-16.

Comment 2.2

Line 16 – authors note that El Nino is a clear example, but they do not reference the pathway or the direction. Clarify.

Response 2.2

After adding a new paragraph to the Introduction in response to Comment 2.4, we decided to merge what were previously the first two paragraphs of the Introduction into a single paragraph. In doing so, we deleted two sentences, including the one to which this comment refers. As a result, this comment no longer applies.

Comment 2.3

Line 28 – “pathogen importation” that occurs at times concurrent with weather conditions suitable for transmission – should be noted. If infected individuals arrive during the time period with very low mosquito density, for example, an imported case may have minimal influence.

Response 2.3

We added text to acknowledge this point on lines 24-25.

Comment 2.4

Line 30 – The authors move from a very general discussion of climate / climate change and population introduction of pathogens to a specific history of an epidemic of dengue in Guangzhou, China. It would be beneficial to have a third paragraph that bridges this more general discussion with some specifics related to dengue and climate change. While some of the references are on dengue in the first two paragraphs, there are an equivalent number on cholera, malaria, etc.

Response 2.4

In response to this suggestion, we added a new paragraph in the Introduction on lines 28-42 that echoes general issues raised in the first paragraph but in a manner that is more specific to dengue. We feel that this has greatly enhanced the Introduction, as we are now able to achieve our original intention of placing our work in a broad context (through what is now the first paragraph) and to introduce the more general reader to salient features of dengue's epidemiology (through this new paragraph).

Comment 2.5

Line 35-37 – Case numbers are fine but not comparable across geographic areas. Since this paragraph is setting up the

differential transmission in SE Asia vs. Guangzhou, I recommend including the cases per 100,000 population. Line 37 and Line 43 – The case numbers from 2014 are variably presented as 47,056 and 37,445. Confirm.

Response 2.5

We apologize for the confusion around the issue of the geographic context of the case numbers described in this passage. Given that we report no information about incidence in Southeast Asia or any other areas outside of China, we feel that it is appropriate to continue our use of case numbers rather than cases per 100,000. We feel that this is particularly appropriate in this context, where case numbers vary across orders of magnitude yet are relatively small in comparison to the overall population of 14 million. Regarding the two different totals for 2014 that the reviewer noted, one reflects the total for all of mainland China, whereas the other reflects the total for the city of Guangzhou, with the latter being the subject of our analysis. In addition to describing the epidemiological situation in Guangzhou specifically, we find it instructive to refer to statistics from mainland China, which are available going back to 1990 and provide a more complete historical context than more recent data specific to Guangzhou. To make this distinction clearer, we reframed our presentation of these geographical contexts such that one paragraph reports only on numbers across the whole of mainland China (lines 44-53) and another paragraph introduces the setting of Guangzhou (beginning on line 55), which is the focus of the manuscript thereafter.

Comment 2.7

Lines 46, 48 – If referring to specific temperature/ humidity/ precip. etc. for 2014, this should be referred to as weather patterns, not climate which typically refers to periods of 20 years or more.

Response 2.7

We revised the text on line 60 and elsewhere accordingly.

Comment 2.8

Lines 44-53 – The review of the articles that have used this dataset prior, indicates an either – or type of result, with some finding that importation is the leading factor and other that “climate” (weather) is the leading factor, however, Cheng et. al. 2016 and Cheng et. al. 2017 both appear to indicate that it is both related to anomalous weather and timing of imported cases. I would suggest revising this paragraph to better reflect the findings from these initial analyses of the data.

Response 2.8

We updated lines 60-63 to better reflect the conclusions from those studies as summarized in Table S1.

Methods:

Comment 2.9

Line 322 – Dengue case definitions are deferred to another paper, but including a one sentence synopsis is important. Are these suspected or lab confirmed?

Response 2.9

We added a one-sentence synopsis of dengue case definitions on lines 398-400.

Comment 2.10

Line 391 – The rationale for the 1-49 day range is not clear. How was the full range of serial intervals determined?

Response 2.10

We clarified our rationale for this choice on lines 442-447 and lines 483-486. In summary, we chose 1-49 days as this spanned the full range of serial intervals that we assumed were possible based on the DENV-specific serial interval formulation using a probability density function derived by Siraj et al.⁴⁸.

Comment 2.11

Line 448-450 – More details on the choice of specific beta distribution parameters for probability of being infected would be useful.

Response 2.11

This step in the likelihood calculation did not actually involve any choices about parameter values. Rather, the beta distribution provides a flexible way to approximate the distribution of simulated incidence (which can only be obtained through a limited number of simulations) with a continuous distribution. Due to a certain mathematical compatibility between beta and binomial distributions (i.e., the conjugate relationship referred to in the text), this step lends itself to straightforward translation of a distribution of simulated outcomes into a likelihood for a certain number of outcomes among a certain number of individuals in such a way that uncertainty in the probability of the outcome is accounted for explicitly. We have enhanced the text in this section of the Methods on lines 558-570 to better communicate this aspect of our approach.

Comment 2.12

Line 468 – The optimum temperature is indicated as 33.3°C – which seems quite high given results from temperature-survival experiments for Ae. albopictus. It is anticipated that this is not just accounting for the optimum mosquito survival/density but takes into account the EIP which shortens considerably as temperature increases.

Response 2.12

We agree with the reviewer that the optimum temperature of 33.3 °C could be high for *Ae. albopictus*, given the temperature-survival experiments that the reviewer refers to. As the reviewer notes though, EIP is also included in determining this prior, given that the prior represents the assumed relationship between temperature and R_0 , which includes multiple temperature-sensitive vector and pathogen traits. We now state that more clearly in the Methods on lines 489-493. Because this temperature optimum of 33.3 °C refers to the prior distribution, we inspected the posterior distribution to determine what optimum temperature was supported by the data (given that the posterior results from a combination of the data, via the likelihood, and the prior). We found that the optimum temperature implied by the posterior (posterior median: 33.6 °C) was similar to the optimum temperature implied by the prior (prior median: 33.3 °C). We interpret this to indicate that while the prior likely had some influence on the posterior, there was not a strong enough signal in the data to pull the posterior estimate of optimum temperature towards lower values. Instead, the data appear to indicate much greater uncertainty in optimum temperature (posterior 95%: 24.8-36.0 °C) than was implied by the prior (prior 95%: 33.2-33.5 °C). We have now added additional information about this on lines 108-109.

Comment 2.13

Further, it does not appear that the relationship between the EIP and temperature is considered explicitly in the modeling framework. This would alter how the three components of the model interact.

Response 2.13

We have now clarified our interpretation of how our model could be considered to implicitly account for the relationship between EIP and temperature that the reviewer refers to. This appears on lines 489-493 in the Methods: “Although not represented explicitly, the $s_T(T_{t-\tau}, \tau)$ function is sufficiently flexible to account for the combined effects of temperature-sensitive virus and vector traits, such as the extrinsic incubation period, mosquito mortality, and mosquito biting. Whereas other models represent those factors explicitly based on mechanistic assumptions, we model their combined influence in a more statistical fashion.”

Results:

Comment 2.14

Lines 90-95: Authors describe differences in the contribution of short, intermediate, and long lag times. It would be useful to define these time periods in terms of days. Suggest color gradation in Fig. 2B to differentiate these more clearly.

Response 2.14

On lines 112-113, we added parenthetical descriptions of the number of days after short, intermediate, and long lag times. We likewise heeded the reviewer’s suggestion to modify Figure 2 to differentiate the days more clearly and shifted the axis perspective of the surface. We also explored the reviewer’s suggestion about a color gradation, but ultimately decided that the shifting of the axis was a simpler and sufficiently effective solution.

Comment 2.15

Lines 120-124: Authors discuss the relative influence of temperature and mosquito density. In the discussion the authors make mention of the importance of temperature on mosquito longevity as a potential explanation of relationships found (lines 265-266). Humidity also influences longevity and could be another potential variable included if there is significant differences (Schmidt et. al. Parasites and Vectors 2018). Another very important parameter that is not discussed is the significant relationship between temperature and the extrinsic incubation period of the virus. This is a critical parameter in determining transmission potential and should receive some attention within the manuscript both in the modeling approach and the interpretation of findings.

Response 2.15

Regarding humidity, in response to this comment and Comment 1.1, we considered four alternative sets of covariates (Supplementary Text 2), two of which included relative humidity as a covariate in the model. After fitting and evaluating these alternative models, we concluded that our original model outperformed alternative models that included relative humidity and precipitation. Regarding EIP, we have now clarified on lines 489-493 in the Methods how our model could be considered to implicitly account for the relationship between EIP and temperature. We also discuss on lines 444-447 in the Methods how EIP is used to derive the description of the serial interval used to determine the time between one generation of human cases and the next. We also draw attention to the importance of EIP in the Introduction on lines 31-33 and in the Discussion on lines 328-331. Finally, to address the potential role of these or other covariates not accounted

for in our model, we added an acknowledgement on lines 366-372 in the Discussion that covariates beyond those used in our analysis could potentially help explain additional inter-annual variation in dengue incidence.

Comment 2.16

Line 171: What is meant by “local conditions”, is this just the mosquito and temperature components of the model?

Response 2.16

We clarified on lines 202-204 that we refer to local conditions as the combination of mosquito density, temperature, and residual variation in local transmission captured by $\beta_0(t)$. The decision to include the latter in this grouping is discussed in greater detail in Response 1.2.

Comment 2.17

Line 177: Why were 2008 importation conditions used as the comparator across the years? Was it highest? Lowest?

Response 2.17

2008 did have the fewest number of imported cases. As indicated on lines 206-211, simulations from these years (2014 local conditions and 2008 importation conditions) were chosen to highlight two extremes to demonstrate the reasoning behind our two-way analysis of variance. This is now stated more clearly following the addition of text on lines 211-214 and in the caption for Figure 6.

Discussion:

Comment 2.18

Authors discuss very briefly the fact that the serogroup that year may have been more infectious and/or pathogenic leading to a greater portion of cases being reported. It would be helpful if any information on serogroup in the region over time could be added and discussed. It is fairly well established that DENV 2 is related to more severe illness. If this was the serotype circulating it would support their argument.

Response 2.18

We have elaborated on our discussion of possible serotype effects in the Discussion on lines 307-314. This now more directly addresses the possibility of an increased probability of symptomatic disease due to either a secondary infection or inherent differences among serotypes. We also make clearer the likelihood of these possibilities based on available evidence from the study area. Regarding potential differences in transmissibility, we have likewise elaborated on our discussion of this topic on lines 326-331 but focused on differences among genotypes rather than serotypes. We feel that the former is more intriguing given recent findings by Fontaine et al.³⁹, which showed that variation in EIP among DENV genotypes within the same serotype could be sufficient to result in epidemics with appreciable differences in size, all else equal. This result is particularly relevant in light of evidence that multiple genotypes were in circulation during the large epidemic experienced in Guangzhou in 2014.

Comment 2.19

Further serotype is a critical component of interannual dengue transmission variability and even if susceptibility is high for all serogroups due to lower levels of historical transmission, sending communities (i.e. those that are the sources of imported transmission) that have high levels of transmission each year would potentially result in greater numbers of imported cases to seed the transmission Guangzhou in multiple geographic areas. The proportion of infected individuals that are traveling into the area that get captured in the surveillance system is likely quite low, though the proportion would not necessarily be expected to vary annually. It is also possible that even if the number of imported cases is not driving the epidemic, spatial patterns of imported cases could further explain some of the high transmission in 2014. It is likely that only a small portion of imported cases are detected.

Response 2.19

We agree with the reviewer's comment that there could be aspects of serotype dynamics in sending communities that affect the proportion of imported infections that are visible to surveillance. We now acknowledge this possibility in the Discussion on lines 331-336. Regarding spatial differences within Guangzhou, we lack spatially resolved data about any factors that could be varying spatially—e.g., mosquito density, population immunity—that might otherwise provide insight about whether spatial patterns of imported cases could help explain the high incidence observed in 2014. Moreover, given that abnormally high local transmission was also observed in prefectures in Guangdong province beyond Guangzhou, we suspect that finer scale spatial variation may not be a significant driver of the epidemic in 2014. We have, however, discussed the potential role of unobserved infections on lines 331-336 and 361-362 and 366.

Comment 2.20

Even if serogroup information in the region is spotty, including some evidence about circulating serotypes would be helpful. Since line 325 indicates that there is sequencing conducted on viral isolates, it would seem this data is available

somewhere to enrich the discussion.

Response 2.20

Just after describing the case data in the Methods, we added a summary on lines 413-420 of what is known from other studies about the serotype and genotype composition of DENV circulating in our study region during the time period of our analysis. This information is also referred to in new text on lines 307-311 and 326-331 in the Discussion.

Comment 2.21

Another potentially interesting factor is the role of imported mosquitoes. Clearly this is not possible to include in an analysis but if transmission across the region was high in 2014, there is a possibility of introduction of virus through infected mosquitoes through importation of goods. Inclusion of information on the overall transmission dynamics regionally would enhance the discussion.

Response 2.21

In response to this comment, we briefly acknowledged this possibility on lines 361-362 in the Discussion. Beyond that, we do not feel that further addressing the issue of DENV importation via infected mosquitoes should be a priority. Although importation of goods can be an important factor for introduction of mosquitoes, we suspect that it is rare for infectious mosquitoes to be imported. There are many factors that make this unlikely, including the chance of becoming infected in the first place, the chance of surviving long enough to become infectious, and for the timing of those events to unfold in such a way that a mosquito is infected at its origin at a young age just before or during its journey to its destination. Once there, it is unlikely that an imported infectious mosquito would live long enough to take more than one or two infectious blood meals, at most. On the other hand, we have direct knowledge of hundreds of cases of imported dengue virus in humans into our study area, and there were likely hundreds, if not thousands, more that went undetected. When an infectious human arrives at its destination, the number of times that person may transmit the virus is limited only by the number of adult female mosquitoes in its surroundings. A recent paper by several of us (Lai et al.⁴², now referenced on line 361) found that human mobility patterns from dengue-endemic countries in Southeast Asia explain a significant amount of variation in overall DENV introduction into this area. On balance, we suspect that importation of infected mosquitoes is very unlikely to play an important role in driving DENV transmission in our study area.

REVIEWERS' COMMENTS:

Reviewer #1 (Remarks to the Author):

The authors have adequately considered and addressed my comments. The main text has been significantly improved and both Supplementary Text 1 and 2 are very useful additions.

I have the following suggestions for the new Supplementary sections:

Supplementary Text 1:

Write down the equation for both the alternative 'simple' model and the primary model (to show how one is a simpler version of the other). Demonstrate which features of the primary model improve the fit compared to the simple model (build up the primary model step by step). Include a figure, following Figs ST2: 15-18 from Supplementary Text 2, to show the difference between the simple model and the primary model (including sub models of the primary model with increasing levels of complexity if appropriate).

The authors state in their rebuttal letter "These same elements of our more complex model were captured by this simpler model in a single line of R code and are, therefore, consistent with our interpretation of the rationale for the reviewer's comment." It would be useful to include this line of code and the corresponding R code for the primary model.

Supplementary Text 2:

When defining each alternative model, include the left-hand side of the equation, e.g. what term is 'm(t)+T+...' equal to? Otherwise, describe which part of the TSIR model this collection of terms represent.

Reviewer #2 (Remarks to the Author):

I appreciate the authors detailed and thorough response to the my comments on the manuscript. I have just a few additional minor comments.

1. In the new second paragraph there is still use of climatic variables but they appear to be referring to weather patterns, not climatic patterns which would be more likely used when comparing geographic areas not temporal variability between years.
2. While I understand that the term "local conditions" is used in part due to the uncertainty in what they capture, I still finding it a vague and unsatisfying term used in the first paragraph of the discussion, perhaps in the first instance a (temperature, mosquito density, and unknown local conditions) could be used. Just something that puts a bit more specificity to what "local conditions" actually encompass.
3. Just a side note - It is possible that precipitation does not come out as an important factor as its primary pathway of influence would likely be on mosquito density which is already accounted for within the model.

REVIEWERS' COMMENTS:

Reviewer #1 (Remarks to the Author):

The authors have adequately considered and addressed my comments. The main text has been significantly improved and both Supplementary Text 1 and 2 are very useful additions.

I have the following suggestions for the new Supplementary sections:

Supplementary Text 1:

Comment 1.1

Write down the equation for both the alternative 'simple' model and the primary model (to show how one is a simpler version of the other). Demonstrate which features of the primary model improve the fit compared to the simple model (build up the primary model step by step).

Response 1.1

This is a good suggestion that the reviewer makes. In response to this comment and Comment 1.3, we included the R code for the simple model in the text. In response to building up the primary model step by step from the simple model, we cannot actually do that as the simple model is not nested within the primary. Instead the simple model As the simple model is not actually a model nested within the primary model, we cannot build up the primary model step by step as there are a number of structural differences between the model.

Comment 1.2

Include a figure, following Figs ST2:15-18 from Supplementary Text 2, to show the difference between the simple model and the primary model (including sub models of the primary model with increasing levels of complexity if appropriate).

Response 1.2

The simple model we use in Supplemental Text 1 (now Supplementary Methods 1) is a deterministic model unlike the primary model we use in the main text. Therefore, we cannot do stochastic simulations like we do with the models (Supplemental Text 2, now Supplementary Methods 2) derived from our primary model. When we plot the model fit in Supplementary Methods 1 Fig. 1, the 95% CI bands we show reflect parameter uncertainty and not stochasticity within the epidemic generating process. This is a one-step ahead model fit, and not a model simulation only driven by imported cases, like we do with the primary model. Because the fit of this simple model to the data to which it was fit is so poor, we do not think it is necessary to extrapolate the model to data to which it was not fit (i.e. simulating model driven by imported cases).

Comment 1.3

The authors state in their rebuttal letter "These same elements of our more complex model were captured by this simpler model in a single line of R code and are, therefore, consistent with our interpretation of the rationale for the reviewer's comment." It would be useful to include this line of code and the corresponding R code for the primary model.

Response 1.3

We included a text box in Supplemental Text 1 (now Supplementary Methods 1) with the R code for the simple model. Ultimately, we decided not to include the corresponding R code for the primary model alongside, as this model is much longer than one line of code. Instead, we created a GitHub repository with extensively commented code, where the code for the primary model is available.

Supplementary Text 2:

Comment 1.4

When defining each alternative model, include the left-hand side of the equation, e.g. what term is 'm(t)+T+...' equal to? Otherwise, describe which part of the TSIR model this collection of terms represent.

Response 1.4

We updated Supplemental Text 2 (now Supplementary Methods 2) to include the left hand side of the equation for each alternative model. Additionally, we included the equation for the TSIR model and indicated how these alternative models would specifically fit into the TSIR model.

Reviewer #2 (Remarks to the Author):

I appreciate the authors detailed and thorough response to the my comments on the manuscript. I have just a few additional minor comments.

Comment 2.1

1. In the new second paragraph there is still use of climatic variables but they appear to be referring to weather patterns, not climatic patterns which would be more likley used when comparing geographic areas not temporal variability between years.

Response 2.1

We updated the text so lines 38, 39, and 42 to say 'weather' conditions instead of climatic conditions.

Comment 2.2

2. While I understand that the term "local conditions" is used in part due to the uncertainty in what they capture, I still finding it a vague and unsatisfying term used in the first paragraph of the discussion, perhaps in the first instance a (temperature, mosquito density, and unknown local conditions) could be used. Just something that puts a bit more specificity to what "local conditions" actually encompass.

Response 2.2

We updated the text in the first paragraph of the discussion (lines 311-312) to include "(temperature, mosquito density, and other unknown local factors)" when we first summarize our approach and results.

Comment 2.3

3. Just a side note - It is possible that precipitation does not come out as an important factor as its primary pathway of influence would likely be on mosquito density which is already accounted for within the model.

Response 2.3

We agree with this point that the reviewer makes regarding the association between precipitation and mosquito density. We make this point at the end of the text in Supplementary Methods 2, and suggest that although precipitation and humidity are meaningful biologically, that the influence of these two variables is mediated through their effects on mosquito density.